# A 10–year climatology of globally distributed ice cloud properties inferred from the CALIPSO observations

Honglin Pan[a], Xinghua Yang[a,*], Kanike Raghavendra Kumar[b,c,**], Ali Mamtimin[a], Minzhong Wang[a], Chenglong Zhou[a], Fan Yang[a], Wen Huo[a], Chaofan Li[d], Jiantao Zhang[a], Lu Meng[a]

[a]*Taklimakan Desert Meteorology Field Experiment Station of CMA, Institute of Desert Meteorology, China Meteorological Administration (CMA), Urumqi 830002, Xinjiang, China*

[b]*Department of Physics, School of Sciences and Humanities, Koneru Lakshmaiah Education Foundation, K. L. University, Green Fields, Vaddeswaram 522502, Guntur, Andhra Pradesh, India*

[c]*Collaborative Innovation Center on Forecast and Evaluation of Meteorological Disaster, Key Laboratory for Aerosol-Cloud-Precipitation of China Meteorological Administration, School of Atmospheric Physics, Nanjing University of Information Science and Technology, Nanjing 210044, Jiangsu, China*

[d]*Collaborative Innovation Center on Forecast and Evaluation of Meteorological Disaster, School of Geographic Sciences, Nanjing University of Information Science and Technology, Nanjing 210044, China*

***Corresponding authors**

Email: yangxh@idm.cn (X. Yang)
kanike.kumar@gmail.com; rkkanike@kluniversity.in (K.R. Kumar)

## ABSTRACT

The present study aims to analyze the climatology of spatiotemporal and vertical distribution characteristics of ice clouds, including the ice cloud fraction (ICF), ice water content (IWC), and ice cloud optical depth (ICOD) for three ice cloud categories (sub-visual, thin, and opaque). Newly released level 3 ice cloud data observed from the Cloud-Aerosol Lidar and Infrared Pathfinder Satellite Observation (CALIPSO) instrument is used in this study for the period 2007-2016. The results revealed that the global means of ICF and IWC were found to be ~10% and ~0.0017g/m$^3$, respectively. On the other hand, the latitude-and-altitude mean distributions of ICF and IWC were found unimodal in all the seasons. During summer, the peak in the ice cloud formation occurred over the equatorial region of the northern hemisphere (NH) which extended further to higher altitudes over the NH equator than the southern hemisphere (SH). However, the opposite was observed in the cold season related to the strong convective activities in tropical areas, variation in the distribution of land and ocean between NH and SH, and the seasonal migration of the inter-tropical convergence zone (ITCZ). Furthermore, the ice clouds detected during the nighttime in summer occurred at high frequency over the SH high-latitude regions, owing to the polar stratospheric clouds (PSCs). The occurrence of sub-visual ice clouds (ICOD<0.01) was infrequent in the tropics and below 5% in other regions. Whereas, the opaque ice clouds (0.3≤ICOD<1, ICOD≥1) occurred most frequently in mid-latitude storm-active regions. The relationships between IWC and relative humidity (RH) and temperature (TE) suggested negative and positive correlations in

the nighttime, respectively. However, the relationship between ICOD and the meteorological variables depends on the range of ICOD.

**Keywords:** Ice cloud properties; Spatiotemporal and vertical distributions; Day and night changes; Meteorological variables; CALIPSO.

## 1. Introduction

Ice clouds are the key regulators of global surface temperature and have enormous implications for the Earth's radiative balance, hydrological cycle, atmospheric circulation, and climate (Wylie et al., 2005; Mülmenstädt et al., 2015). On the one hand, ice clouds can not only reflect shortwave radiation but can also absorb some outgoing thermal radiation, contributing to the cooling and warming, respectively of the Earth's atmosphere (Chen et al., 2000; Liou, 1986). Moreover, the net cooling or warming effect of ice clouds depends on their optical and microphysical properties, such as optical depth and water content, as well as on their macro-physical characteristics, such as cloud location and coverage throughout the atmosphere (Hong et al., 2016; Lee et al., 2009; Baran, 2012). On the other hand, owing to the complex interactions between ice clouds and aerosols, the contribution of ice clouds to climate leads to remaining large uncertainties (Zhang et al., 2015; Pan et al., 2019; Jiang et al., 2018; Zhao et al., 2018). Existing studies argue that the climatology of ice clouds obtained from global cloud models (GCMs) presents a

relatively large difference in terms of their spatiotemporal distribution, compared with the data retrieved from satellite measurements (Eliasson et al., 2011; Hong et al., 2016). Consequently, the long-term and high-resolution measurements by both the ground and satellite-based remote sensors are of high importance for developing better ice cloud parameterization, which in turn is expected to improve the accuracy of GCMs.

The vertical distribution of ice clouds plays a pivotal role in determining the radiative forcing of ice clouds. Furthermore, compared with horizontally resolved measurements, vertical measurements of ice cloud properties are insufficient around the globe, owing to the complexity of sampling. Previous studies have analyzed the occurrence frequency of ice clouds as well as their optical and microphysical properties in terms of spatiotemporal variability (King et al., 2013; Holz et al., 2008; Kubar et al., 2009; Sun et al., 2011; Berry et al., 2019; Lauer and Hamilton, 2013). However, owing to the above-mentioned importance and complexity of atmospheric ice clouds, these studies are not sufficient. Moreover, many of these studies have been limited to small spatial regions and short temporal periods, as well as certain classifications of ice clouds. For example, Berry and Mace (2014) investigated whether ice clouds with the ice water path (IWP) of ~20 g/m$^2$ contribute to the obvious heating of Earth during summer monsoons in Asia. Tsushima et al. (2013) found that the error in the frequency of anvil cirrus in the tropics biased the cloud radiative effect. Therefore, long-term and large-scale climatology studies of ice clouds are necessary, to delineate some specific physical processes, which can be

regarded as the cause of the biggest errors in cloud modeling.

While ground-based observations can significantly contribute to temporal coverage, they cannot constitute a global database of ice cloud data and are limited mostly to land areas. On the contrary, satellites can extend ground-based observations to include land and ocean areas, and complement multiple measurable capabilities based on different wavelengths throughout the electromagnetic spectrum (Kumar et al., 2018; Boiyo et al., 2018; Mace and Berry, 2017). Overall, the passive sensors such as the moderate-resolution imaging spectroradiometer (MODIS) can yield column-integrated ice cloud properties such as IWP/LWP (Wang et al., 2016; King et al., 2013; Yang et al., 2007; Oreopoulos et al., 2014). However, the active instrument like the cloud aerosol lidar and infrared pathfinder satellite observations (CALIPSO) joined the "A-Train" satellites, since 2006, providing an unprecedented information about the vertical structure characteristics of ice clouds, such as their ice water content and optical depth (Gao et al., 2014; Jiang et al., 2018; Winker et al., 2010).

In this paper, we utilized CALIPSO level 3 lidar ice cloud data to investigate the vertical distributions of seasonal and diurnal variations, as well as the global geographical distributions of the ice cloud fraction (hereafter, ICF), ice water content (hereafter, IWC), and ice cloud optical depth (hereafter, ICOD) during the recent 10-year observation period of 2007–2016. To examine seasonal climatology, the whole year was considered and divided into four seasons as spring (March, April, May), summer (June, July, August), autumn (September, October, November), and winter (December, January, February). In addition, we analyzed the relationship

between these multiple ice cloud parameters and meteorological conditions. The rest of this paper is organized as follows: Section 2 describes the datasets and methods, Section 3 illustrates the results and discussion, and Section 4 lists the main conclusions of this study.

## 2. Data and methods

Since April 2006, the CALIPSO satellite launched by NASA carries the Cloud-Aerosol Lidar with Orthogonal Polarization (CALIOP) sensor, which is the first nadir-viewing dual-wavelength (i.e., 532 nm and 1064 nm) satellite lidar in a helio-synchronous orbit at a height of 705 km, with a repeat cycle of 16 days and a local equator-crossing times (EXTs) of 13:30 and 01:30 LT (Winker et al., 2007). The outputs of the Level 1 product obtained from the CALIOP serves as 532 nm wavelength parallel-polarized and 532 nm wavelength perpendicular-polarized attenuated backscatter coefficients, respectively. On the other hand, the attenuated backscatter coefficient at 1064 nm, which can produce the level 2 data product given the input data and algorithms. The algorithms include the scene classification algorithm (SCA), which contains a family of algorithms for feature detection (i.e., clouds and aerosols; ice and water clouds), and the hybrid extinction retrieval algorithm (HERA), which retrieves cloud extinction data and infers the cloud optical depth (Young et al., 2008; Liu et al., 2009; Pan et al., 2019). In addition, the level 1 and 2 data from CALIOP have high horizontal resolutions of 333m and 1km for the heights in the range of 0.5–8.2 km and 8.2–20.2 km, respectively, and vertical resolutions of 30 m and 60 m at 532 nm (Hunt et al., 2009). The profiles obtained

from the CALIOP are averaged with an increasing signal to noise ratio (SNR), which allows the measurement of weaker layers (clouds or aerosols) (Vaughan et al., 2009; Pan et al., 2016, 2019).

Recently, the CALIPSO lidar instrument released (on December 2018) level 3 version 1.00 ice cloud monthly gridded data from January 2007 to December 2016, which were then used in the present study. The spatial resolutions were 2° latitude and 2.5° longitude, and with a vertical resolution of 120 m ranged from -0.5 km to 20.2 km above the mean sea level (AMSL), generating three different types profiles (i.e., daytime, nighttime, and both) depending on the light conditions; where the file data were created from level 2 version 4.10 cloud profile products. The primary variables in that dataset were the IWC histogram, sampling counts, and meteorological context. Further, we estimated the ICF in the zonal distribution based on Eq. (1) given below:

$$ICF_{zonal} = \frac{\sum_{long=-178.75°}^{178.75°} ICAS}{\sum_{long=-178.75°}^{178.75°} (CS + CFS)} \quad (1)$$

Here, CS, CFS, ICAS refer to the number of cloud samples, number of cloud-free samples (clear sky or aerosol features), and number of ice cloud-accepted samples, respectively. The latitudinal ICF describes the cloud fraction as a function of latitude and height, which requires integration over longitudinal samples (for brevity denoted as “long”, ranging from -178.75° to 178.75°) for each latitude and height bin.

Besides, we calculated the zonal distribution of the IWC based on Eq. (2) shown

below:

$$\mathrm{IWC_{zonal}} = \frac{\sum_{\mathrm{long}=-178.75^{\circ}}^{178.75^{\circ}} \bullet \left( \sum_{\mathrm{bin}=2}^{\mathrm{bin}=16} \mathrm{IWCBB} \times \mathrm{IWCH} + \sum_{\mathrm{bin}=43}^{\mathrm{bin}=19} \mathrm{IWCBB} \times \mathrm{IWCH} \right)}{\sum_{\mathrm{long}=-178.75^{\circ}}^{178.75^{\circ}} (\mathrm{CS} + \mathrm{CFS})} \qquad (2)$$

Here, IWCBB represents 44 bins of the full distribution of the IWC, and we excluded the small and large outliers in bins 1, 17, 18, and 44. IWCH refers to the histogram of the IWC. Specifically, the cloud occurrence is considered in the denominator. Therefore, the equation derives the grid-averaged IWC. Detailed information about the product can be found online at https://www-calipso.larc.nasa.gov/resources/calipso_users_guide/data_summaries/l3/lid_l3_ice_cloud_v1-00_v01_desc.php

We also selected the level 3 version 1.00 cloud occurrence monthly gridded data product with the longitudinal and latitudinal resolutions of 5° and 2°, respectively, and at an altitude of 60 m; as well as three files including day and nighttime observations, and all observations. The ICOD histogram with seven levels of optical depth was utilized in this dataset. Based on the daytime and nighttime files, the diurnal variations of the ICF, IWC, and ICOD were inspected by analyzing the night-minus-day measurements.

## 3. Results and discussion

### *3.1. Spatial distributions of ICF and IWC*

The spatial distributions of the 10-year mean of ICF and IWC over the globe are

shown in Figs. 1and 2 (up to ~ ±84° latitude owing to the limitation of the CALIPSO view), respectively. In Fig. 1, the main coverage of ice clouds can be found in the vicinity of the equator (~±15° latitude), which reaches ~30% in southeastern Asia, western Africa, South America, as well as ~20% in certain parts of the Pacific Ocean. Over the mid-latitude regions, the occurrence frequency of ice clouds is relatively significant owing to frequent storm activities (Hong et al., 2015). Desert regions located in Northwestern China (Taklimakan desert), Northern Africa (Sahara desert), Southern America, and Central Australia exhibited smaller coverage of ice clouds owing to weak water vapor and convective activities (Pan et al., 2019). On the other hand, high-latitude regions exhibited a comparatively high frequency of ice clouds which can be attributed to the fact that polar stratosphere clouds (PSCs) are captured, owing to the CALIOP sensitivity. Further, the number of ice clouds in the polar area of the Southern Hemisphere (hereafter, SH) was higher than that of the Northern Hemisphere (hereafter, NH), which is consistent with the previous results (Sassen et al., 2008; Huang et al., 2015). The total ICF as the 10-year mean around the globe was estimated as ~10%. Following Fig. 2, the global geographical distribution of IWC is consistent with the corresponding ICF. One exception is that the concentration of the IWC in the polar region of the SH is smaller than NH. The global 10-year averaged value of IWC was found to be ~0.0017g/m$^3$ (Fig. 2).

### *3.2. Seasonal latitude-and-altitude distributions of ICF*

In this section, we discussed the latitude-and-altitude distributions of a 10-year mean of ICF distributed over four seasons, based on Eq. (1). As shown in Fig. 3, the

coverage of ICF generally exhibited a unimodal distribution where the peak is under the "flatness" tropical tropopause altitude (refers to the upper boundary of the troposphere) in the middle part and decreases steadily towards both the polar areas of SH and NH for the entire study period. In the summertime, the maximum ICF of ~40% occurred over the equator of the NH, and ice clouds can reach higher altitudes towards the north of the equator than the south. Meanwhile, the opposite phenomena were observed in the winter period. This is mainly attributed to the strong convective activities in tropical areas, the distributional variation of land and ocean over the NH and SH, and the seasonal migration in the position of the inter-tropical convergence zone (ITCZ). These results are consistent with that reported by Huang et al. (2015) and Su et al. (2008). Also, the PSCs could obviously be observed in the SH during the summer and autumn seasons. Moreover, the availability of the CALIPSO nighttime data during the warm season in the high latitudes of NH is limited as revealed from Fig. 3. The same has been observed and reported by Anderson et al. (2015), owing to the fact that measurements were being limited over the latitudes from 55° to 80°N. It is to be noted that we had used the arithmetic mean to compute the annual distribution of ICF, and extrapolation can be used for more complete and accurate data, which is, however, beyond the scope of the present study. The minima, maxima, and mean of the ICF observed during daytime and nighttime for four seasons over the globe are listed in Table 1.

### *3.3. Seasonal latitude-and-altitude distributions of IWC*

As illustrated in Fig. 4, the latitude-and-altitude distributions of the 10-year

mean CALIOP observations of IWC revealed asymmetrical distribution. However, the measurements of ICF and IWC present contradictory differences during nighttime and daytime and is attributed to the sampling-induced bias. In Fig. 5, the IWC histograms derived from the CALIOP during the study period for four seasons exhibited unimodal patterns, and one evident mode occurs at ~10mg/m$^3$. Furthermore, smaller (larger) IWC values have more (less) samples during the nighttime than daytime, respectively. Meanwhile, the latitude-and-altitude distributions of IWC showed a "spike-shaped structure" at an altitude of ~4 km for all the seasons in both the hemispheres. However, in-depth analysis and studies should be performed to further explore this discrepancy. A detailed summary (including the minima, maxima, and mean) of the IWC data for four seasons is given in Table 2. Noticeably, we excluded the maximum of the IWC presented in Table 2 because the IWC values under 0.01g/m$^3$ accounted for 99% of the data.

### *3.4. Mean profiles of diurnal variations of ICF and IWC*

Based on the daytime and nighttime files of the CALIPSO level 3 data, we used "nighttime minus daytime" measured data to explore the diurnal variations of mean profiles of ICF and IWC for different latitudinal bands. We focused on the summertime data, which exhibited stronger variations between nighttime and daytime, to analyze the diurnal variations of the aforementioned quantities. Moreover, it is important to break down the diurnal variability in terms of real and artificial variabilities (e.g., instrumentation-induced, classification-induced, and sampling-induced variabilities). Owing to the sunlight-related noise in the daytime,

optically thin layers of ice clouds cannot be probed by the CALIOP, compared with the nighttime. Further, the classification into liquid and ice phases of clouds are also affected; that is, stronger noise during the daytime may more negatively affect the classification of cloud types or clouds and aerosols, compared during the nighttime. Consequently, the CALIPSO level 3 daytime-and-nighttime data exhibited significant statistical variations, which can be explained in terms of artificial daily variations. In addition, taking into account that these parameters can be affected by pollution for the bins above the Earth surface (Jiang et al., 2018; Huang et al., 2015; Sassen and Wang, 2008), and for the following analysis, we only sampled data for an altitude of at least 2 km. The results of this analysis are shown in Fig.6, including for the diurnal variations and the overall number of samples of ICF and IWC.

Over the SH tropics (30°~0°S), the diurnal variability of ICF exhibited two peaks at ~10 km and ~15 km. However, a stronger variation of 0.1 over the NH tropics (0°~30°N) was found for the same height. In general, the ICF exhibited higher occurrence frequency for the upper-tropospheric layer of 10–15 km in the tropics (30°S~30°N). Interestingly, the total number of samples for the daytime was more than nighttime which is below ~5 km over the NH tropics (because the x-axis is on the logarithmic scale and hence, the negative values were neglected). Over the SH mid-latitudes (60°~30°S), the diurnal difference between the ICF peaks at ~8 km, and negative values of the ICF observed below ~3 km indicate smaller ICF during the nighttime, as well as the diurnal variability of the total number of samples of ICF peaks at ~18 km. A difference between the daily variations of ICF peaks at ~10 km in

the NH mid-latitudes (30°~60°N) and the negative values of ICF can also be observed below ~5 km. In the high-latitude SH region (90°~60°S], the ICF exhibited a larger variation of 0.1 at ~8 km, owing to the higher occurrence frequency of PSCs. Conversely, in the high-latitude NH region [60°~90°N), the ICF variation at an altitude of ~8 km was smaller than ~0.1, which is attributed to the limited nighttime data collection by CALIOP in the NH high-latitude region.

Also, the diurnal differences in IWC were also analyzed, and the results are shown in Fig. 6. Overall, a larger variation (negative trend) occurred in the NH compared to SH. Over the tropics, the difference between the diurnal variability peaks was ~12 km (approximately -1.3 mg/$m^3$ in the NH). Over the mid-latitude region of the NH, two peaks in the diurnal variability were observed, at ~11 km (approximately -0.8 mg/$m^3$) and ~4 km (approximately -1 mg/$m^3$), respectively, and at ~7 km (approximately -3.5 mg/$m^3$) and ~4 km (approximately -2.5 mg/$m^3$) in the NH high-latitude regions. Over the mid- and high-latitude regions of the SH, the diurnal difference in the IWC exhibited a single peak at a height of ~9 km and ~4 km, respectively. Based on the above results of the diurnal variability of ICF and IWC, we revealed some interesting facts for these observations which are mentioned as follows. First, the CALIOP is more sensitive to detect weak signals from ice clouds, yielding more samples in the nighttime than the daytime, owing to the sunlight during the daytime. Secondly, the IWC has more samples with small values during the nighttime than in the daytime and fewer samples with large values in the nighttime than the daytime, which is following with the analysis and interpretation given in Section 3.3.

This explains the behavior of opposite trends in the diurnal variations of IWC and ICF.

### *3.5. Spatial and seasonal changes of frequency occurrences of ICOD*

The geographical and seasonal averaged frequency of occurrences for the three types of ICOD or six sub-types over the globe is shown in Fig. 7 following the classification proposed by Sassen et al. (1992) and Hong et al. (2016). The three categories of ICOD are namely, sub-visual (ICOD<0.01, 0.01≤ICOD<0.03), thin (0.03≤ICOD<0.10, 0.10≤ICOD<0.30), and opaque (0.30≤ICOD<1, ICOD≥1). Sub-visual ice clouds occur frequently over the tropics and constitute less than 5% elsewhere, except the SH polar region, where sub-visual ice clouds are detected during summer. Besides, the ice clouds with 0.01≤ICOD<0.03 are almost absent over the mid-latitude regions; and in the tropics and high-latitude regions, they occur with higher frequency. Thin ice clouds occur more frequently on the ocean than land. On the other hand, their concentration is higher in the tropics compared to other regions. Ice clouds with ICOD≥1 occur with low frequency over the tropics, but with higher frequency in the mid-latitude region following active storms in this region (Hong et al., 2015).

Also, Table 3 summarizes the minimal, maximal, and mean occurrence frequencies of ICOD for the six sub-types during the four seasons over the globe. Overall, the six groups of ICOD exhibited small seasonal variations concerning their mean occurrence frequencies. One exception is with the ICOD ≥1 in the warm season, for which the mean occurrence is ~26% which is smaller than the other three seasons.

Meanwhile, the maxima for different ICOD were found larger during summer than in the other seasons and accounted for 37%, 42%, 72%, 42%, 63%, and 75%, respectively for the six sub-types.

### *3.6. Diurnal variations of frequency occurrences of ICOD*

To characterize the zonal profiles of occurrence frequency of the ICOD for the above-mentioned six groups, we quantified the occurrence frequency as the ratio of number of ICOD samples in a certain category to the overall number of samples (sum of the six sub-groups of ICOD after data screening); and this quantification was performed for each vertical layer. In the following study, we only considered the warm season to illustrate the diurnal variability of each ICOD owing to the aforementioned results, with obvious seasonal variations. It is observed from Fig. 8 that the number of samples in each of the ICOD categories revealed that the CALIOP probes more ice cloud samples during nights compared to days. However, one exception is that for all the ICOD categories over the NH high-latitude region where a number of samples were acquired during days than nights (because the x-axis is on the logarithmic scale, negative values were not considered), which can be interpreted as a limitation on the CALIOP nighttime data availability over the region. Notably, the diurnal differences in the vertical profiles for all the ICOD samples were nearly overlapped above ~15 km, except over the SH high-latitude region.

In general, the occurrence frequency increased as the value of all ICOD increased at an altitude less than 15 km. On the contrary, the occurrence frequency was inversely proportional to the value of all ICOD above the cutoff shown in the

first and second panels of Fig. 8. We further observed that the diurnal differences of mean zonal profiles of ICOD depend on the latitude. Over the NH tropics, ice clouds with ICOD<0.01 and 0.01≤ICOD<0.03 were found less and more frequent, respectively above ~18 km. Thin ice clouds exhibited small diurnal variations at all altitudes. Ice clouds with 0.30≤ICOD<1 were found less which is common at altitudes between 4 km and 6 km during nighttime. For ICOD≥1, the diurnal difference exhibited two peaks at ~3km and ~5km, with higher frequency in the nighttime. Over the SH tropics, the amplitude of the diurnal difference was smaller than that observed over the NH tropics; moreover, ice clouds with 0.01≤ICOD<0.10 exhibited an opposite trend of the diurnal difference, compared with the NH tropics. In the NH mid-latitude region, the ice clouds with ICOD<0.10 were found less frequent during nighttime, and two peaks were observed at heights of ~16 km and ~19 km. However, the ice clouds with 0.01≤ICOD<0.30 exhibited higher (lower) occurrence frequency for altitudes from 15 km (17 km) to 17 km (20 km) during nighttime. Also, the ice clouds with 0.03≤ICOD<0.10 were more frequent at altitudes from ~18 km to ~20 km during nighttime; and whereas, ice clouds with 0.10≤ICOD<0.30 were found less frequent below ~7 km in the night. The diurnal variability of clouds with 0.30≤ICOD<1 exhibited a bimodal pattern (smaller nighttime frequency) for altitudes from ~3 km to ~7 km, as well as for altitudes from ~7 km to ~15 km. For clouds with ICOD≥1, the diurnal difference in the frequency of occurrence exhibited two peaks for the altitudes of ~2 km and ~5 km. Similarly, the amplitude of the daily difference in the SH mid-latitude region was smaller than in

the NH. Over the SH high-latitude region, ice clouds generally exhibited little diurnal difference at nearly all levels, except for 0.03≤ICOD<0.10 above ~12 km, and ICOD≥1 below ~3 km; especially, clouds with ICOD≤0.01 were found more frequent above an altitude of ~10 km. Over the NH high-latitude region, the diurnal differences for all ICOD classes were found less frequent during nighttime, and the sub-visual ice clouds illustrated that daily differences were approximately 2–3 times stronger than for the thin and opaque ice clouds at higher altitudes (above ~12 km). Besides the limitation on the CALIPSO data acquired during nighttime in the NH high-latitude region, we need to consider instrumentation-induced differences. The CALIPSO orbits Earth in an helio-synchronous orbit at a height of 705 km, with local EXTs of 13:30 and 01:30. Consequently, the CALIOP measurements are performed in the early afternoon and after midnight, contributing to the sampling bias between midday and midnight hours (Stephens et al., 2002; Winker et al., 2009; Huang et al., 2013; Pan et al., 2019). Moreover, a large body of research has verified that daily maxima of deep convection activity (which can transport ice clouds to higher altitudes) and precipitation prevail in late afternoons or early evenings (Nesbitt et al., 2003; Khain et al., 2005). Therefore, a higher occurrence of ice clouds (especially optically thin ice clouds) can be probed by the CALIOP during nighttime rather than daytime.

### *3.7. Relationship between meteorological conditions and IWC, and ICOD*

The results obtained and elaborated in previous sections mapped the climatology of seasonal and geographical distributions of IWC and ICOD for six sub-groups, and

their diurnal differences observed in the vertical profiles of different zonal bands. Here, we studied the relationship between meteorological conditions and microphysical and optical properties of ice clouds. Moreover, we focused on analyzing the summertime data only, and two meteorological parameters were utilized namely, relative humidity (hereafter, RH) and temperature (hereafter, TE); the values of these parameters were obtained from the CALIPSO platform.

Fig. 9 shows the 10-year global distribution of contour density plots drawn between the IWC, RH, and TE during nighttime and daytime. In general, data count (N) is relatively scarce during the night than day, attributed to the limited data collection by the CALIOP during nighttime. We also studied the relationship between IWC and RH, which found positively correlated, with the poor correlation coefficient (r) of 0.23 during the daytime. It is also revealed that the values of IWC peak in the range 0.9–2.7mg/$m^3$, for a comparatively low RH of approximately 36–39%. During the nighttime, the IWC and RH exhibited moderate correlation (r = 0.43), and the IWC peaked in the range of 0.28–1.4 mg/$m^3$ when the RH varied between 36 and 42%. Furthermore, the data points used for the relationship between IWC and RH tended to exhibit a larger spread for nighttime compared to daytime. This can be interpreted as higher noise owing to the background sunlight during daytime compared to nighttime; as a result, fewer samples could be acquired by the CALIOP during the daytime. Also, deep convection activities occurred more frequently in the night than day, allowing to transport many more ice clouds (especially for little ice crystals) into high altitude, and broadening the range of IWC and RH values for night

measurements. The TE and IWC were negatively correlated (r = -0.11) during the daytime, and IWC peaks in the range 0.0–1.98mg/m$^3$ for the cold (-34°C~ -32°C) TE. At night, the IWC and TE were more negatively correlated (r= -0.48), and the IWC peaks at 0.0–1.4mg/m$^3$, with the TE in the range of -34°Cto -32°C. Likewise, the data count of IWC/TE tended to attain higher/lower values during nighttime compared with the daytime.

Next, we analyzed the relationship between the meteorological parameters (RH and TE) and the occurrence frequency of ICOD for the six sub-categories. The results are summarized in the contour density plots shown in Fig. 10. Overall, the ICOD for the six groups peaked basically in the same range of RH and TE, either during daytime or nighttime. However, the magnitudes were different for day and nighttime, and the data points were more dispersed for the nighttime. This can be attributed to ice clouds that are mainly formed in the upper layers of the troposphere, and to the fact that stronger noise-induced a sampling bias in the daytime measurements. This can also be attributed to more frequent convective activity and precipitation during nighttime. In addition, the correlation between the occurrences for different ICOD and RH during nighttime was higher than daytime, except for the 0.03≤ICOD<0.10 and 0.30≤ICOD<1 group. Meanwhile, the TE and occurrence frequency of ICOD also presented a similar correlation. One exception was that ice clouds with ICOD≥1 exhibited a smaller correlation coefficient during nighttime than daytime. Moreover, the association between RH and ICOD<0.01 was the strongest, with the correlation coefficient of 0.39. Also, the association between TE and ICOD<0.01 was found

strong, with a negative correlation coefficient of -0.30 during nighttime, compared with the other studied ICOD groups.

## 4. Summary and conclusions

In this study, we conducted a statistical analysis to understand the climatology of global ice cloud properties including ICF, IWC, and ICOD with six sub-categories using 10-year long-term (2007–2016) measurements observed by the CALIPSO. Firstly, the geographical distribution of the global 10-year averaged ICF was found ~10%. The main coverage of ice clouds is in the vicinity of the equator, which takes up ~30% of Southeastern Asia, Western Africa, and South America, and ~20% of the Pacific Ocean. Over the mid-latitude regions, the occurrence frequency of ice clouds was relatively high, owing to frequent storm activities. For the desert regions, such as Northwestern China (Taklimakan Desert), Northern Africa (Sahara Desert), Southern America, and Central Australia, the ice cloud coverage was smaller. For the SH high-latitude region, the frequency of ice clouds was relatively high, which can be attributed to the selective capture of PSCs, owing to the sensitivity specifications of the CALIOP. Additionally, the spatial distribution of the IWC was largely consistent with that of the ICF, and the global 10-year average of the IWC was ~0.0017g/m$^3$.

The seasonal latitude-and-altitude distributions of ICF generally exhibited a unimodal distribution, in which peak values occurred at the “flatness” tropical tropopause altitude in the middle part, and decreased steadily toward the two sides (polar areas) in both hemispheres during the study period for all seasons. Moreover, we found the global 10-year mean of nighttime data (including IWC and ICF)

collected by the CALIPSO during the summertime suffers from limited data availability for high-latitude regions in the NH. Meanwhile, the 10-year averaged ICF has a maximum of more than ~40% for tropical and SH polar areas in summer. The vertical distributions of 10-year mean IWC exhibited a "spike-shaped structure" at the altitude of ~4 km in all seasons and both hemispheres.

Also, the diurnal difference of ICF exhibited two peaks at ~10 km and ~15 km in the tropical zone. Over the SH and NH mid-latitude regions, the discrepancy occurred in the peaks at ~8 km and ~10 km, respectively. Negative values of daily variation of the ICF occurred in the NH high-latitude region at the height of ~8 km, owing to the restrictions on the data utilization during the nighttime. The magnitudes of the diurnal difference of the IWC are larger in the NH than SH, and negative IWC was observed for all of the considered altitudes. And the occurrence frequency increased as the ICOD increased for altitudes under ~15 km, and the occurrence frequency was inversely proportional to the ICOD value above the cut-off.

Further, the relationships between meteorological conditions and IWC, and ICOD also were investigated. The IWC peaked in the range of 0.9-2.7 $mg/m^3$ (0.28–1.4 $mg/m^3$) for relatively low RH of 36–39% (36–42%) during the daytime (nighttime), with the correlation coefficient of 0.43. For TE and IWC, the correlation coefficient (r = -0.48) was more negative during the nighttime than daytime, and IWC peaked between 0 and 1.98 $mg/m^3$ (0–1.4 $mg/m^3$) for the cold TE (-34°~ -32°), during the daytime (nighttime). All of the ICOD peaks are basically in the same range of RH and TE values, either for the daytime or nighttime. However, the magnitudes

for daytime and nighttime are different. The strongest association is between RH (TE) and ICOD<0.01, with the correlation coefficient of 0.39 (-0.3) during the nighttime.

In general, our analysis using the level 3 version 1.0 profile product indicates that spatiotemporal and vertical distributions of the ice cloud properties are comparatively reasonable and reliable in most of the regions around the globe. In the future, more in-depth optimization of the CALIPSO retrieval algorithms and quality control algorithms should be conducted.

**Funding Sources**

This work was financially supported by the Strategic Priority Research Program of the Chinese Academy of Sciences (Grant No. XDA20100306), the National Natural Science Foundation of China (Grant Nos. 41775030, 41875019, 41575008), and Flexible Talents Introducing Project of Xinjiang for the year 2017-2018.

**Acknowledgments**

We are also grateful to the CALIPSO (https://eosweb.larc.nasa.gov/) instrument scientific teams at NASA for the provision of satellite data, which is available online and formed the main database in the present work.

**Conflicts of interest**

The authors declare that they have no competing financial interests or personal relationships that could have appeared to influence the work reported in this article.

**Author contributions:**

All the authors contributed to shaping the ideas and reviewing the paper.

**Honglin Pan**: Conceptualization, methodology, formal analysis, visualization, original draft-writing.

**Xinghua Yang**: Conceptualization, funding acquisition, project administration, resources, supervision, review & editing.

**Kanike Raghavendra Kumar**: Methodology, resources, supervision, original draft preparation, review & editing.

**Ali Mamtimin**: Data curation, software, visualization.

**Minzhong Wang**: Resources.

**Chenglong Zhou**: Formal analysis, review & editing.

**Fan Yang**: Investigation.

**Wen Huo**: Data curation.

**Chaofan Li**: Data curation.

**Jiantao Zhang**: Investigation.

**Lu Meng**: Investigation.

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

**Table 1.** The minimum, maximum, and mean of ICF observed during daytime and nighttime for four seasons over the globe between 2007 and 2016.

| Season | daytime | | | nighttime | | |
|---|---|---|---|---|---|---|
| | minimum | maximum | mean | minimum | maximum | mean |
| Spring | <1% | 35% | 6% | <1% | 37% | 7% |
| Summer | **<1%** | **31%** | **5%** | **<1%** | **48%** | **8%** |
| Autumn | <1% | 32% | 5% | <1% | 38% | 7% |
| Winter | <1% | 25% | 5% | <1% | 37% | 7% |
| Annual | <1% | 31% | 5% | <1% | 40% | 7% |

**Table 2.** Same as in Table 1, but for IWC. The unit of IWC is g/m$^3$.

| Season | Daytime | | | Nighttime | | |
|---|---|---|---|---|---|---|
| | minimum | maximum | mean | minimum | maximum | mean |
| Spring | <0.00001 | 0.00660 | 0.00120 | <0.00001 | 0.00890 | 0.00096 |
| Summer | **<0.00001** | **0.00740** | **0.00110** | **<0.00001** | **0.01530** | **0.00087** |
| Autumn | <0.00001 | 0.01300 | 0.00110 | <0.00001 | 0.01230 | 0.00095 |
| Winter | <0.00001 | 0.00550 | 0.00110 | <0.00001 | 0.00560 | 0.00099 |
| Annual | <0.00001 | 0.00810 | 0.00110 | <0.00001 | 0.01050 | 0.00094 |

**Table 3.**Same as Table 1, but for ICOD. The min and max represents minimum and maximum, respectively.

| Season | <0.01 | | | [0.01,0.03) | | | [0.03,0.10) | | | [0.10,0.30) | | | [0.30,1.00) | | | ≥1.00 | | |
|---|---|---|---|---|---|---|---|---|---|---|---|---|---|---|---|---|---|---|
| | Min | Max | Avg | Min | Max | Avg | Min | Max | Avg | Min | Max | Avg | Min | Max | Avg | Min | Max | Avg |
| Spring | <0.01 | 0.19 | 0.03 | 0.01 | 0.29 | 0.07 | 0.05 | 0.32 | 0.14 | 0.11 | 0.37 | 0.20 | 0.12 | 0.44 | 0.28 | 0.02 | 0.52 | 0.29 |
| Summer | <0.01 | 0.37 | 0.04 | <0.01 | 0.42 | 0.08 | 0.01 | 0.72 | 0.15 | 0.02 | 0.42 | 0.20 | 0.10 | 0.63 | 0.27 | <0.01 | 0.75 | 0.26 |
| Autumn | <0.01 | 0.26 | 0.03 | 0.01 | 0.37 | 0.07 | 0.05 | 0.49 | 0.14 | 0.09 | 0.50 | 0.20 | 0.02 | 0.48 | 0.28 | <0.01 | 0.58 | 0.28 |
| Winter | <0.01 | 0.32 | 0.03 | 0.01 | 0.34 | 0.07 | 0.04 | 0.45 | 0.14 | 0.09 | 0.38 | 0.20 | 0.02 | 0.48 | 0.28 | 0.001 | 0.62 | 0.29 |

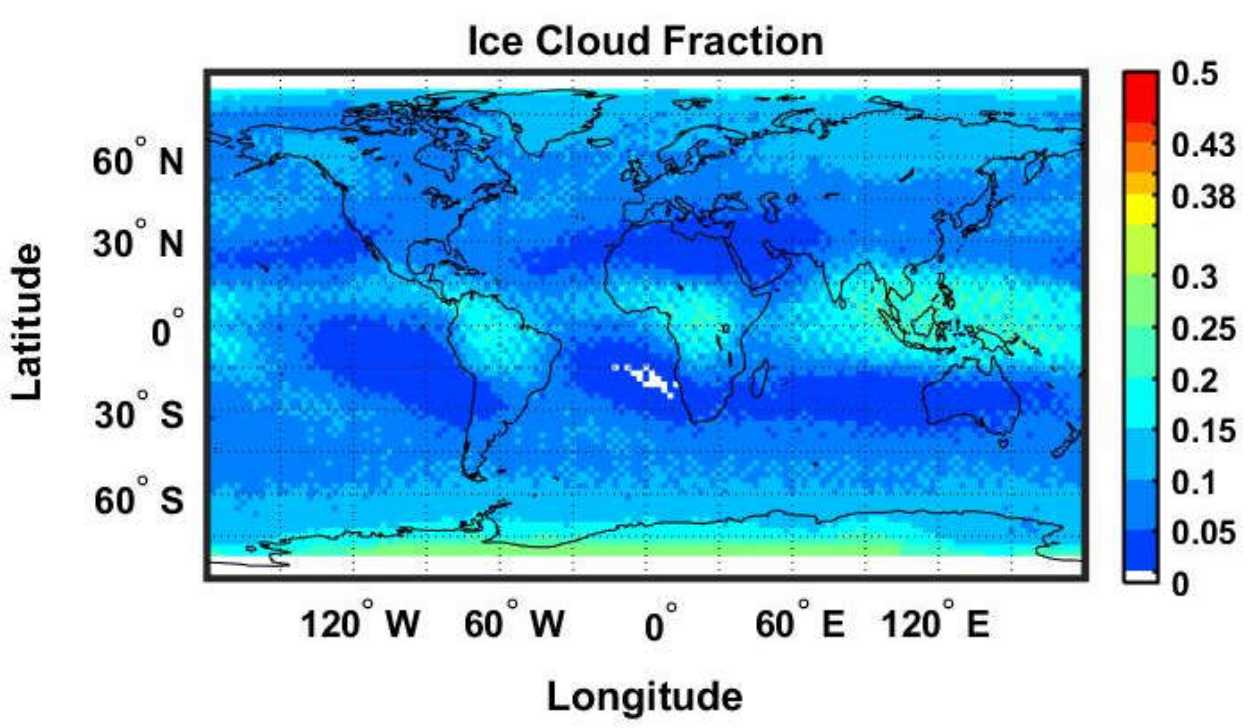

**Fig. 1.** Geographical distribution of a 10-year mean of ICF retrieved from the CALIOP measurements (day plus night condition). The white color represents the value less than 0.01.

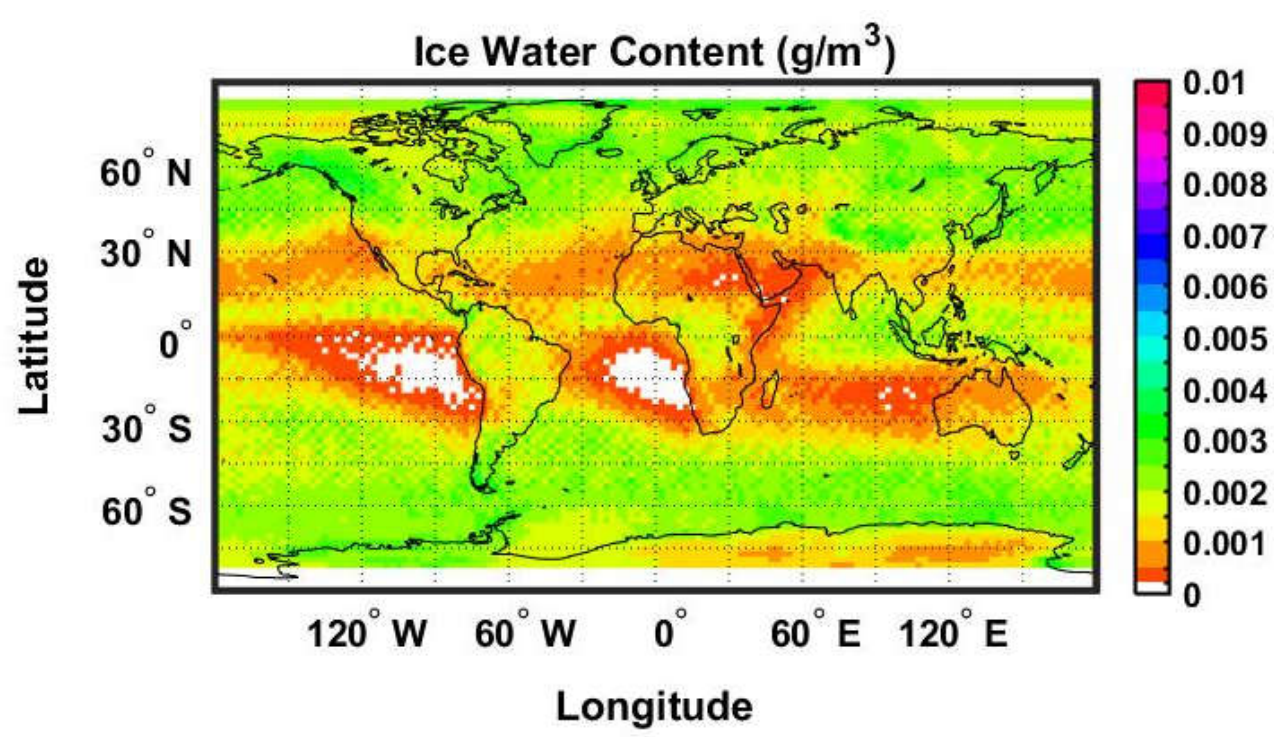

**Fig. 2.** Same as in Fig.1, but for the IWC. The white color represents the value less than 0.0002g/m$^3$.

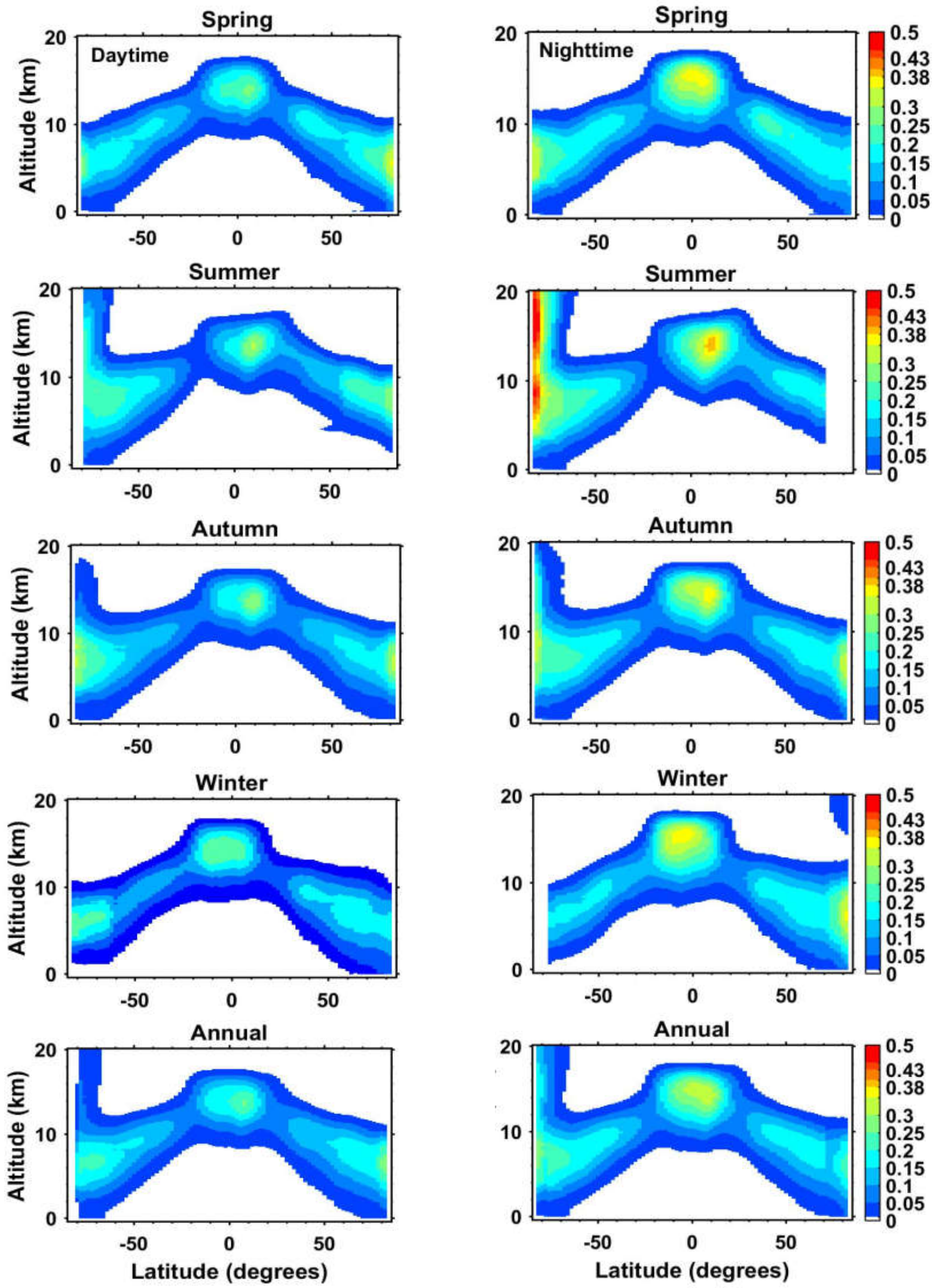

**Fig. 3.** Seasonal and annual distributions of the 10-year mean ICF over latitude and altitude during the daytime (left panels) and nighttime (right panels) for four seasons observed from the CALIOP. The white color represents the value less than 0.01.

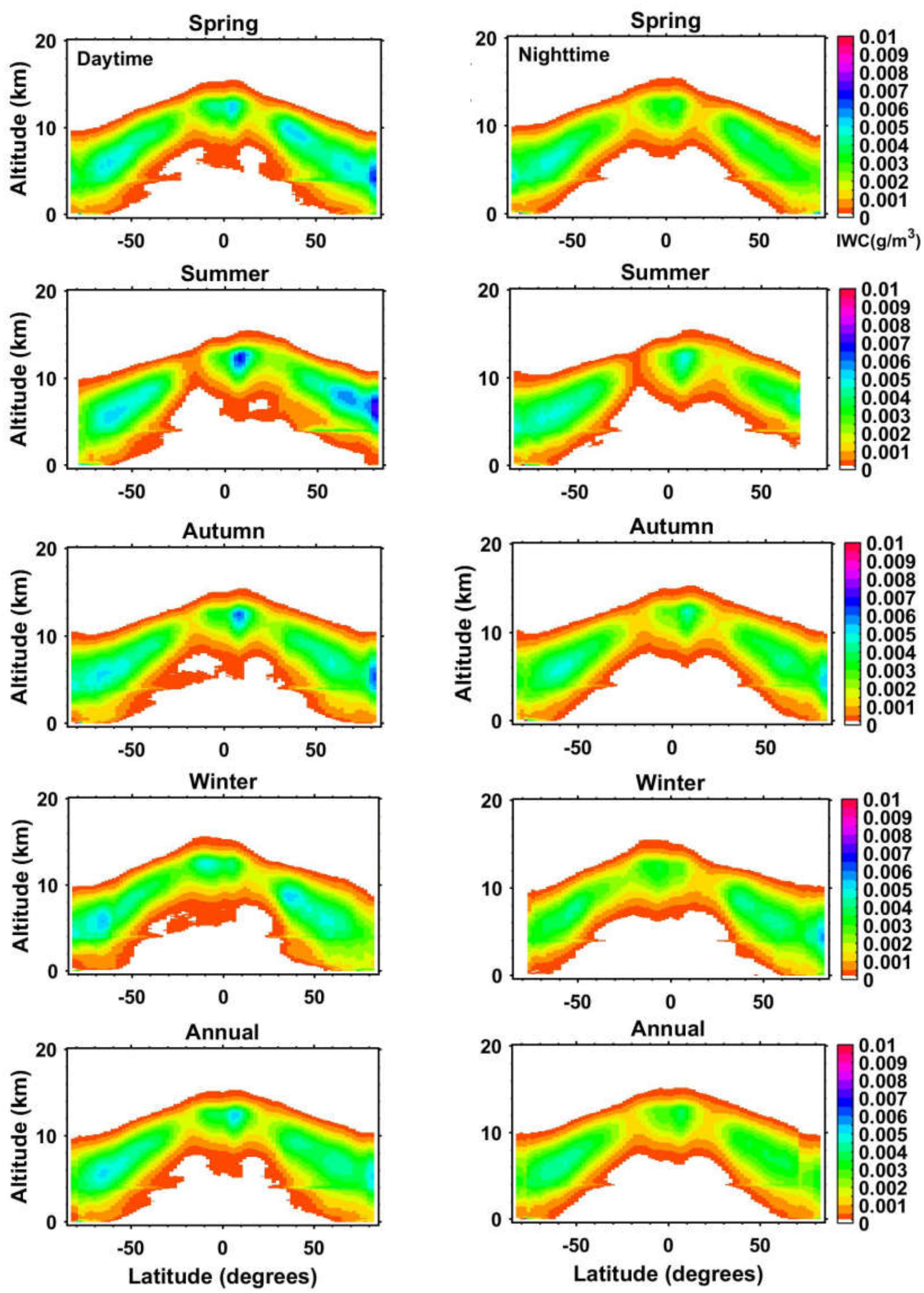

**Fig. 4.** Same as in Fig.3, but for the IWC. The white color represents the value less than 0.0002 g/m$^3$.

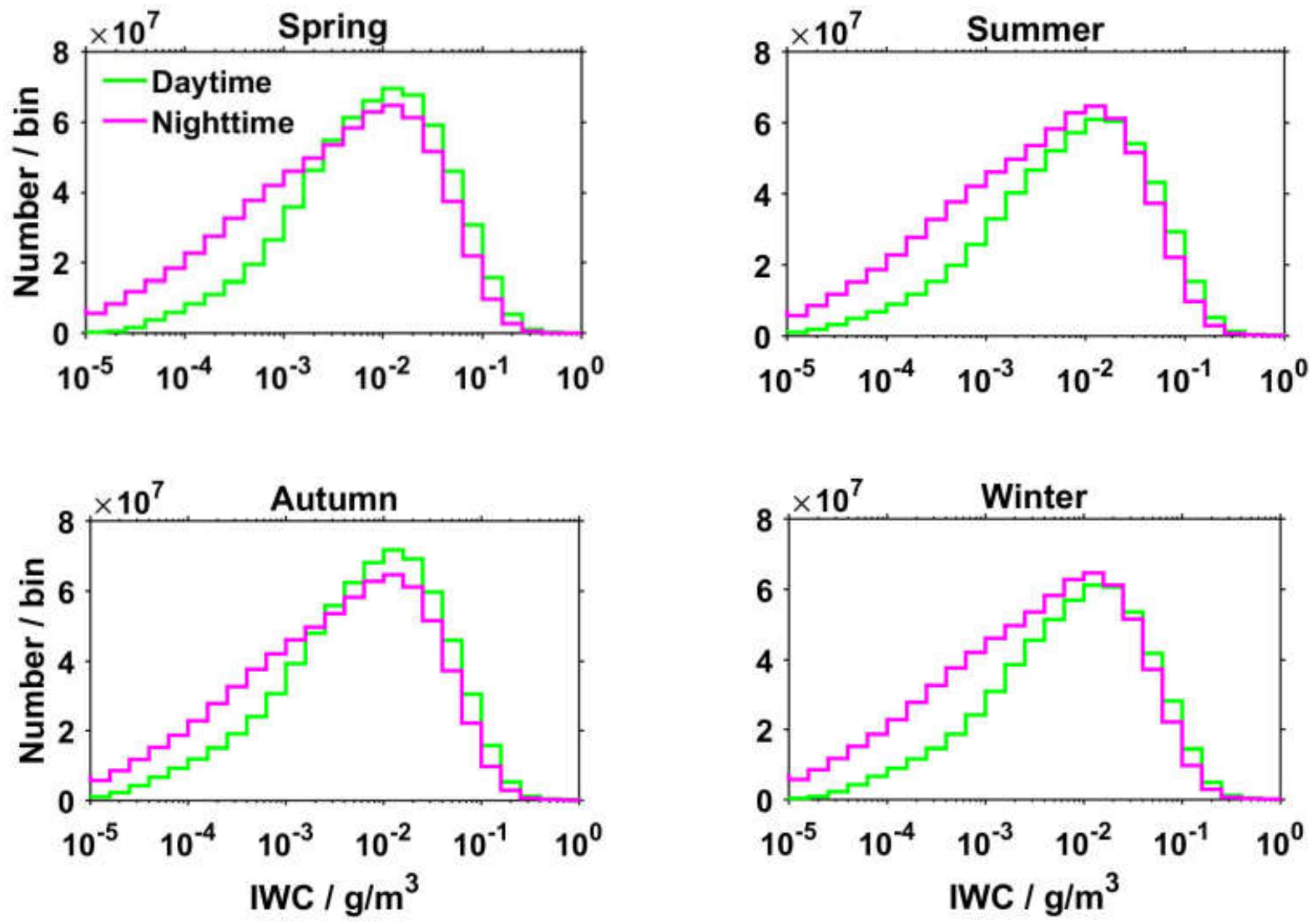

**Fig. 5.** Histograms of IWC derived from the 10-year measurements of the CALIOP for four seasons.

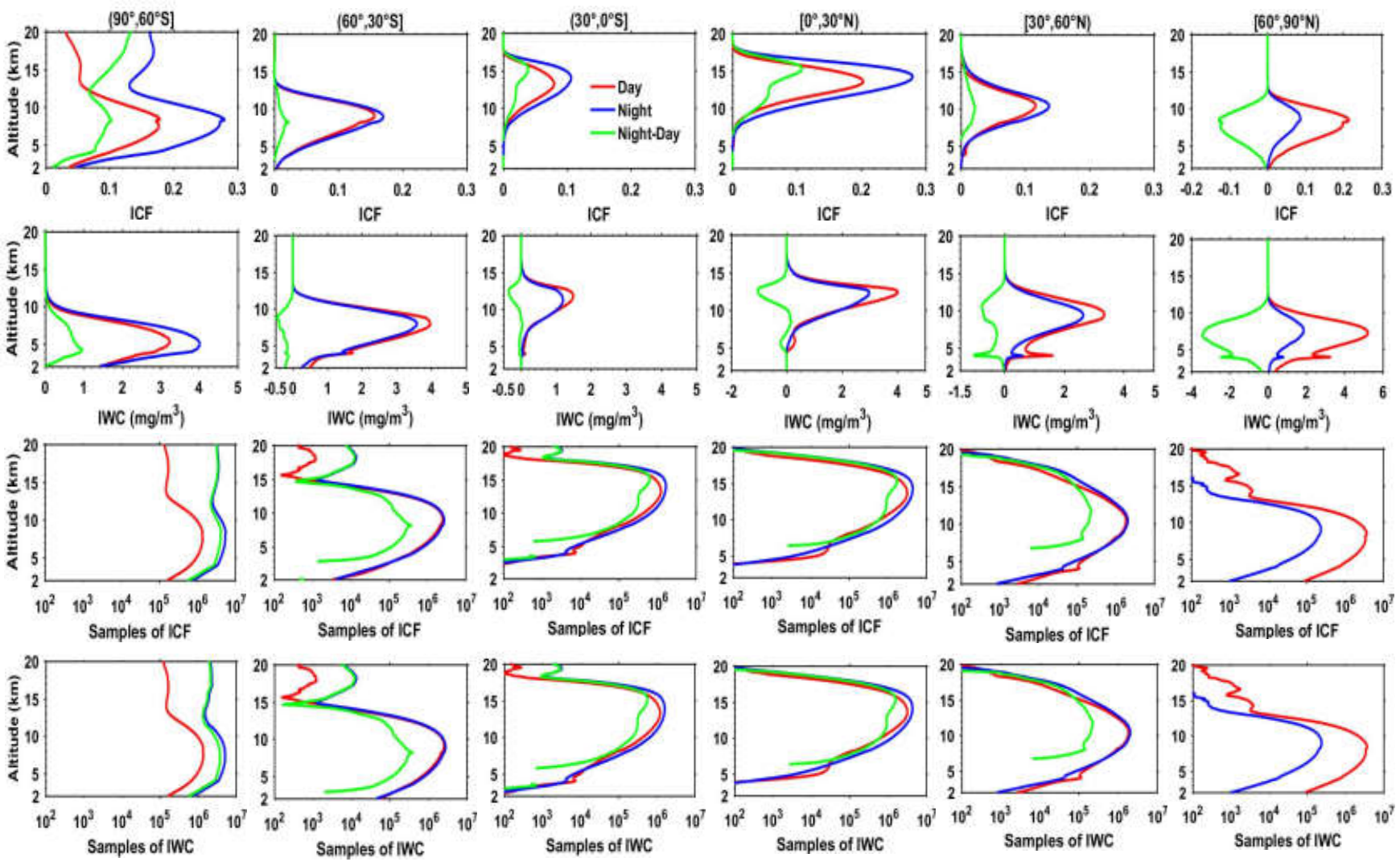

**Fig. 6.** Diurnal variations (night-minus-day measurements) of zonal mean profiles of frequency occurrences of ICF and IWC (the first and second row), vertical profiles of a 10-year total number of ICF and IWC samples (the third and fourth rows).

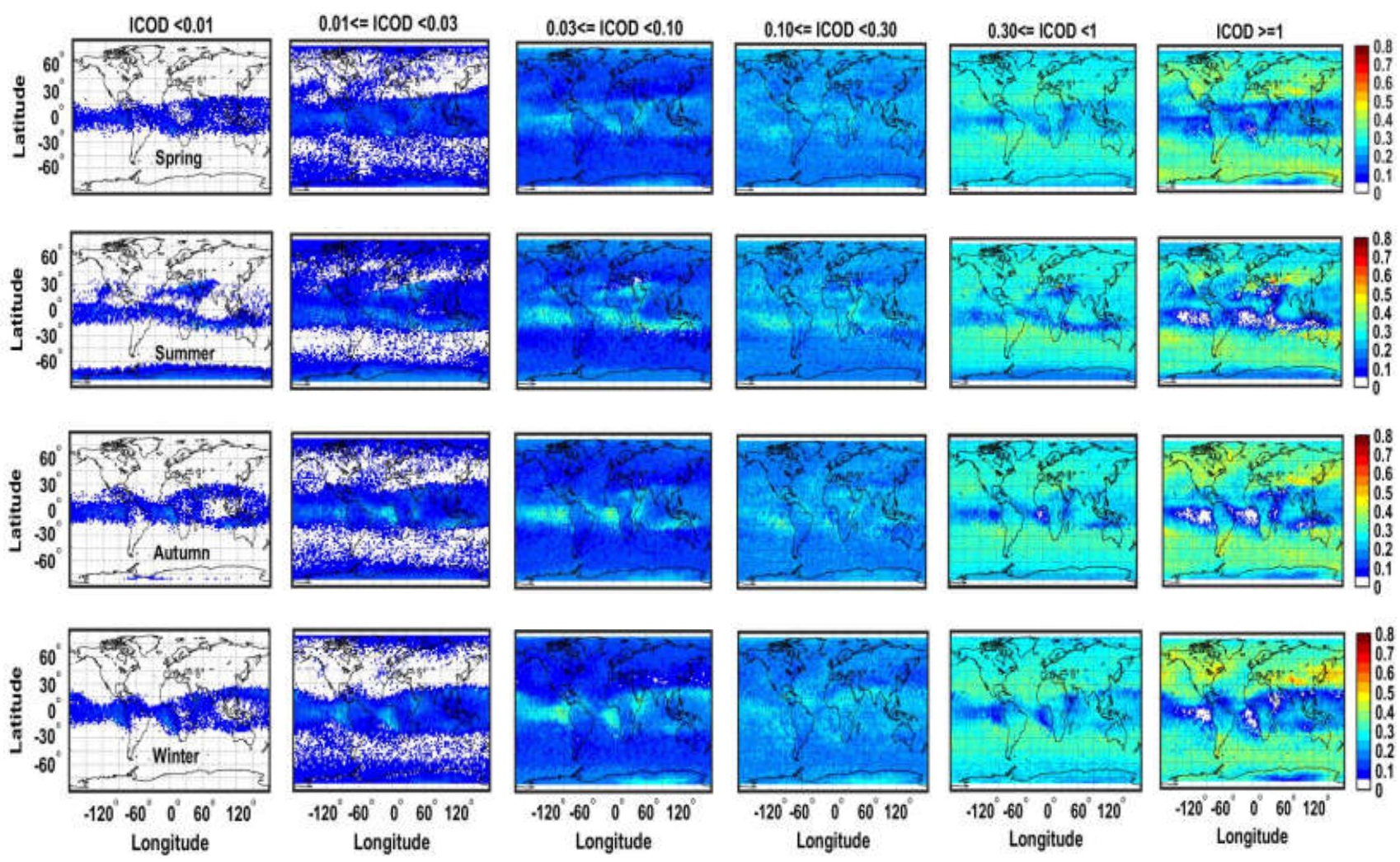

**Fig. 7.** Spatial and seasonal changes of frequency occurrences of ICOD over six ranges based on the 10-year measurements of the CALIOP (day plus night). The white color represents value less than 0.05.

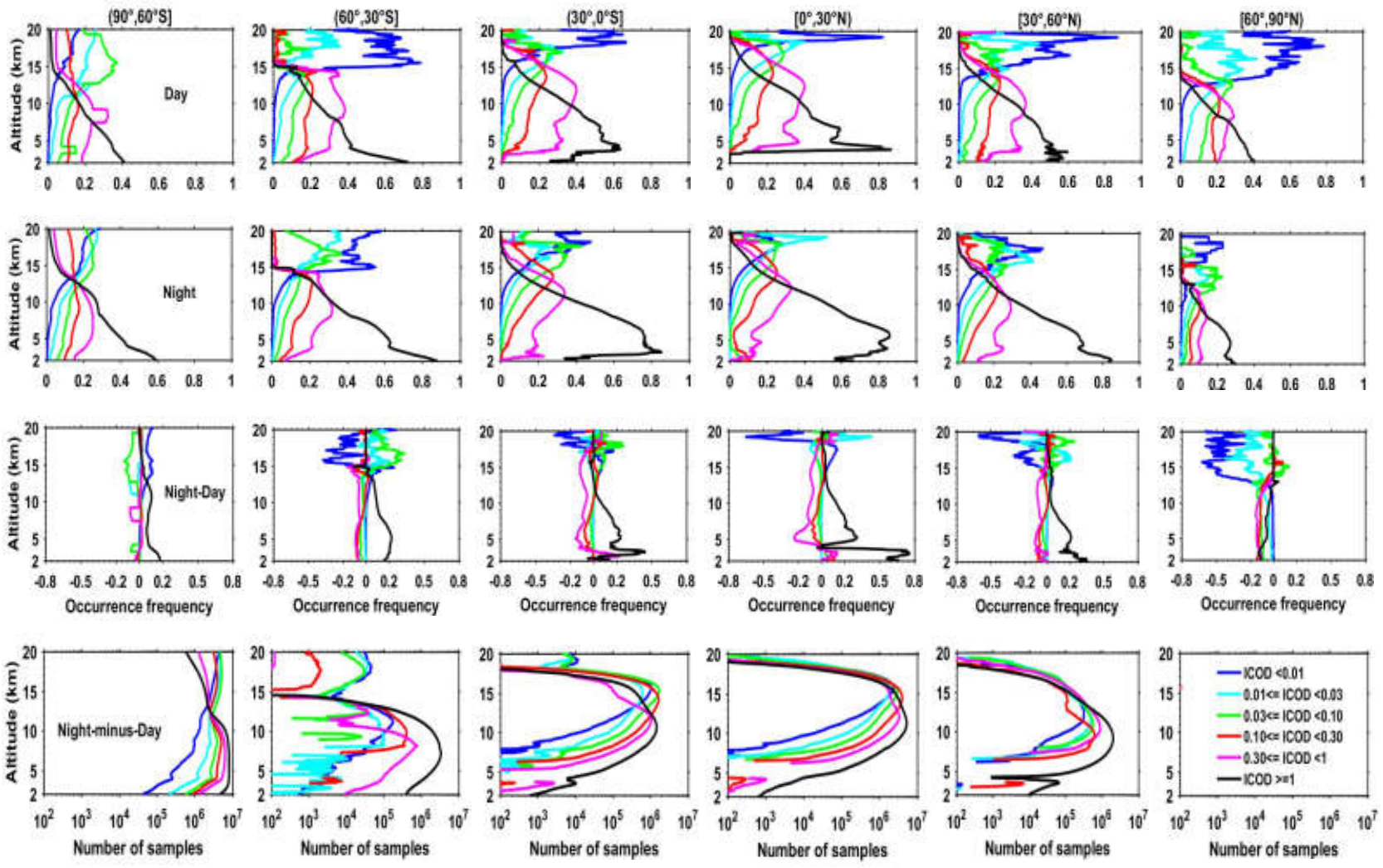

**Fig. 8.** Diurnal (day and night) variations of zonal mean profiles of frequency of occurrences of ICOD over six ranges (the first and second rows), diurnal variation (night-minus-day measurements) of: occurrence frequency profiles of ICOD over six groups (the third row), vertical profiles obtained from the 10-year CALIOP measurements of total number of ICOD samples for six groups (the fourth row).

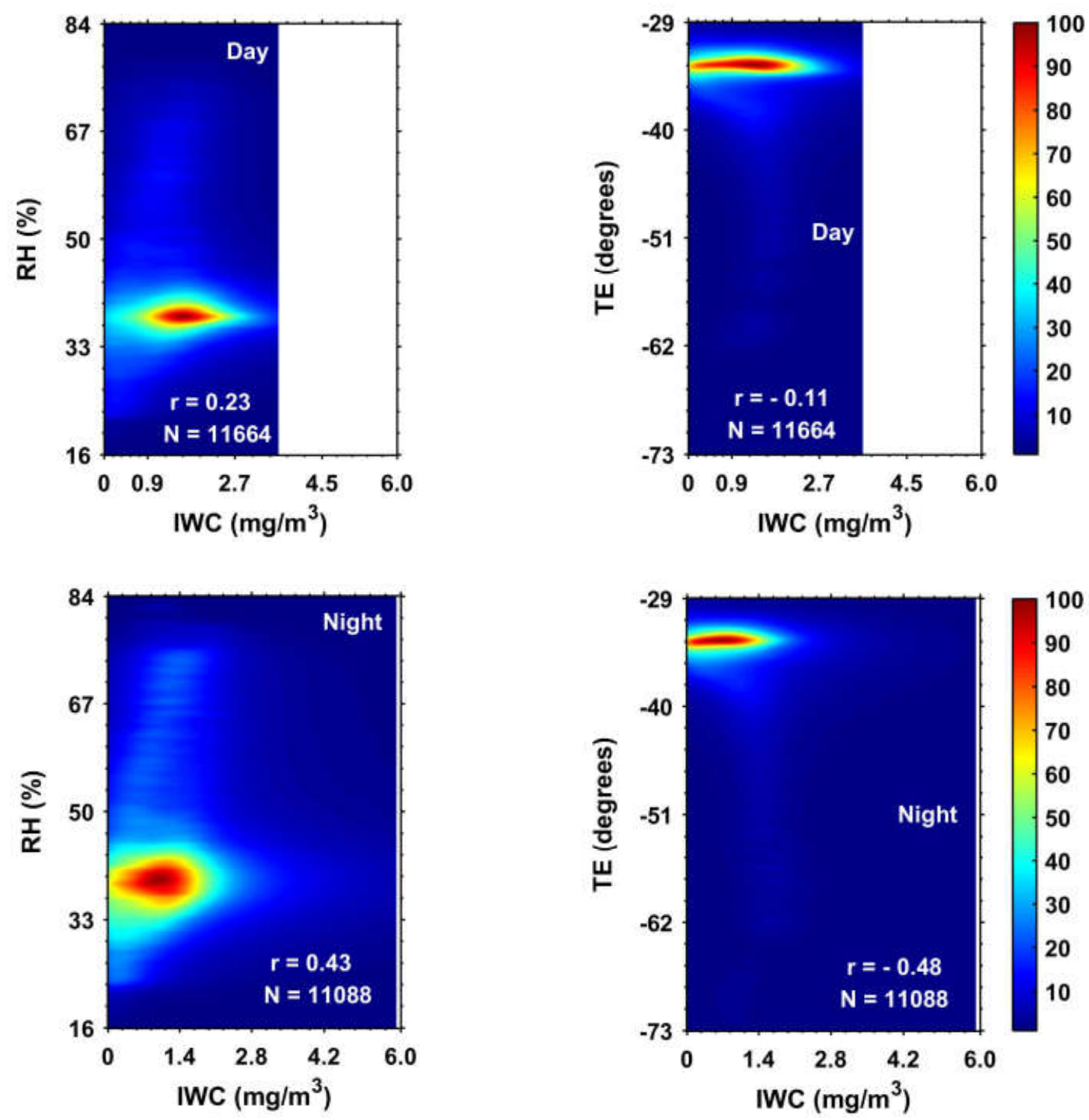

**Fig. 9.** The relationships between the averaged IWC and RH (left column) and TE (right column) during the daytime and nighttime over the globe based on the 10-year measurements of the CALIOP.

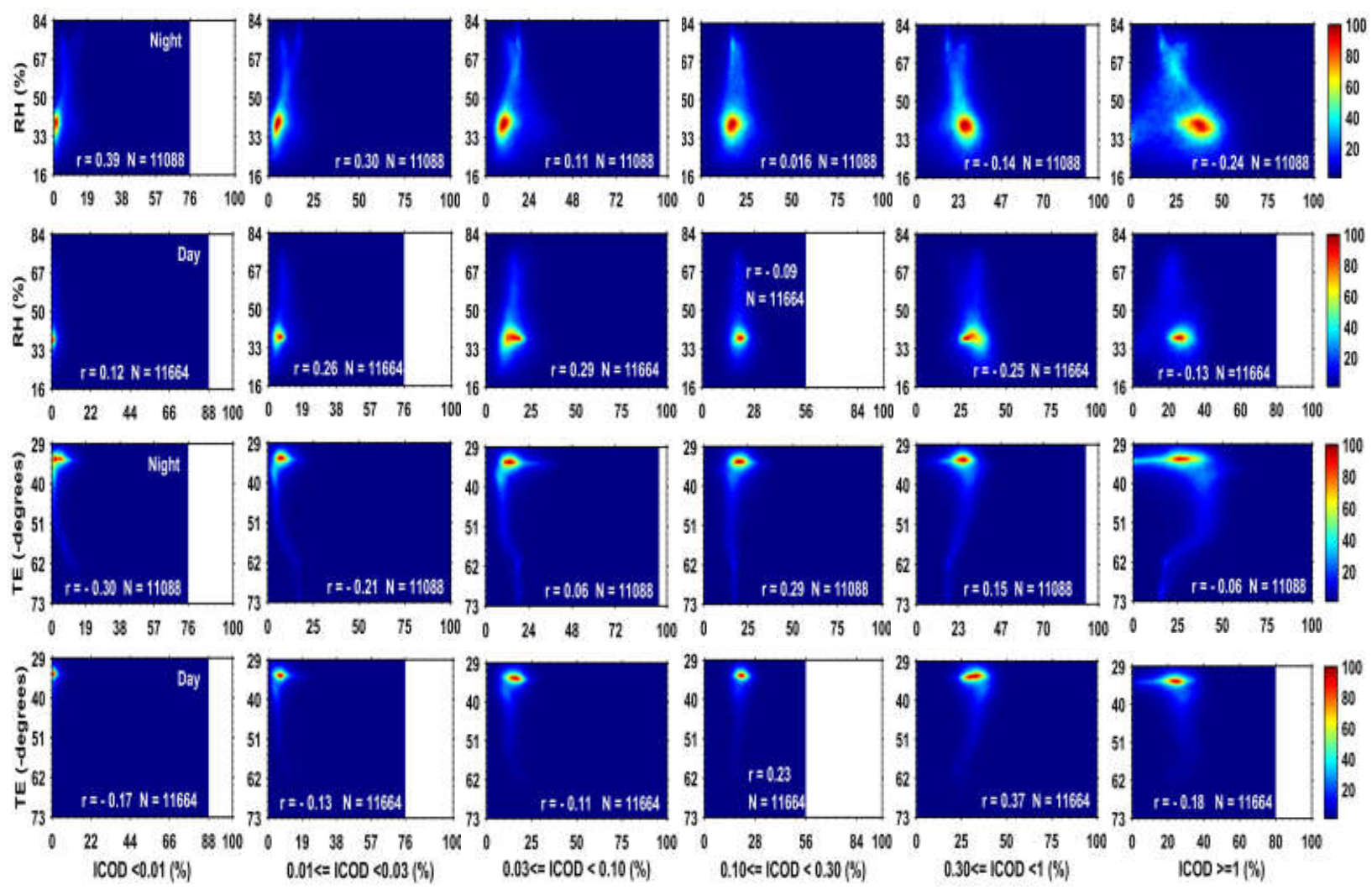

**Fig. 10.** The relationships between occurrence frequency of different ICOD and averaged RH (first and second rows) and TE (third and fourth rows) during nighttime and daytime over the globe derived from a 10-year measurement of the CALIOP. The unit of ICOD in x axis is the percentage.