# Peer review of "A 10–year climatology of globally distributed ice cloud properties inferred from the CALIPSO observations"

_Atmospheric Chemistry and Physics, 2019_

## Referee Comment (RC1) · Anonymous Referee #1 · 4 Mar 2020

In this paper, the authors consider recent CALIPSO level 3 ice cloud detections and IWC retrievals, monthly gridded on a lon-lat-z grid over the 2007-2016 period. They average these values along various dimensions (altitude and longitude, mostly) in each season, plot and describe the results. They also consider the seasonal variations of the geographic distribution of ice clouds of various optical depths, and the diurnal variation of the altitude distribution of ice clouds of various optical depths.

As far as I can tell, the authors have correctly averaged and plotted the variables present in the level 3 files, and thus the article provides a good documentation of the dataset. The authors however did not in my opinion have clear scientific questions in

mind when they drew up this study, and this shows in their methodological choices : 1) unless I'm mistaken the authors built the ICF maps from a 3D gridded cloud dataset, 2) the authors include PSCs in their ice cloud statistics. While these approaches are not technically incorrect, they make it impossible to compare the results presented here with existing literature, in which CF maps are generally built by considering cloudy and non-cloudy profiles in specific altitude ranges, and where tropospheric and stratospheric clouds are considered separately. The second half of the paper includes a puzzling comparison of IWC vs. RH and Temperature, which is never justified and brings little insight. Since the paper does not go far beyond a description of the figures, and there is no discussion of the results, the scientific value of the paper is not obvious. I expand on these points and others below.

Major comments

My first major comment concerns the approach followed by the authors to build a representation of the dataset they study. This representation makes it hard to put their results into perspective by comparing them with existing literature. As the authors do not do this themselves, it is not possible to evaluate if the work done by the authors has any scientific significance. In practice, I mean that as far as I can tell (this is not made clear in the paper) the authors choose to average ice cloud detections over entire vertical profiles (between -0.5km and 20km ASL) to build maps of ice cloud fraction. This choice means that low tropospheric clouds, high tropospheric clouds and stratospheric clouds are all mixed together in the resulting maps. Since those three kinds of clouds are driven by very different atmospheric processes and have very different impacts on the climate system, mixing them all together means it is not possible to derive learning from the maps presented here. Discussion of these maps is limited to e.g. pointing out that large CF over the poles can be linked to the presence of stratospheric clouds. Meanwhile, since the level 3 data used as starting point is altitude-sensitive, it would be entirely possible to separate ice clouds based on their altitude and build ICF maps for low, mid, and high-level clouds. Mixing all the vertical levels together in maps also

means that PSCs are included in the ICF in polar regions. PSCs are not necessarily ice : they can be made of NAT crystals that include HNO3 or STS droplets that include H2SO4. It is unclear to me if the stratospheric "ice cloud detections" really only include ice PSCs or if they include all the other PSCs, which are more frequent but should not qualify as ice clouds. In any case, the processes that drive the distribution of PSCs are completely different from the ones that drive tropospheric ice clouds, and both kinds of clouds cannot be mixed together if we want to understand things about them. My suggestion to the authors would be 1) to eliminate PSCs from the dataset before creating any statistics, by using monthly maps of tropopause altitudes based on e.g. ERA5, and 2) to create maps of ICF for different vertical levels and discuss those separately (see below). Altitude levels to separate low/mid/high clouds can be found in e.g. Chepfer et al., 2010. doi: 10.1029/2009JD012251

Section 3.7 (Fig. 9 and 10) tries to identify relationships between ice water content and relative humidity and temperature. The authors never justify was this is a good idea.In the end, they find very poor correlations (always less than 0.5); these rather argue that there is no reason for these variables to be correlated and thus no reason to investigate their relationship. For the time being, this section and the attached figures could be summed up as "Our investigations did not suggest any significant correlation between IWC and either RH or Temperature". I recommend that the authors either explain why it is important/useful to investigate these relationships, and discuss how their results bring something interesting, or drop the whole section altogether.

Finally, the scientific content of the paper is very limited. The discussion of results is limited to explaining sampling variations due to instrumental biases (e.g. the poor sampling of nighttime data during the polar summer, whose obvious explanation is never provided, see below). I would expect this paper to try to learn things about ice clouds, not about how CALIPSO sees them. Once CALIPSO limitations and specific sampling behaviour are taken into account, what can we say with confidence about the remaining variations of ice cloud fractions and IWC? This should be discussed in

the paper, and put in perspective with existing results from the literature. It is telling that beyond section 3.1 almost no work is cited, because no perspective is provided to understand why the results are important. The summary and conclusion section cites zero reference. I recommend that the authors attempt to put their results into perspective by comparing them with existing studies describing the global distribution of ice clouds and IWC, maybe using other instruments or reanalyses.

In the end, it is unclear to me wether the article wants to study cloud climatologies (as the abstract suggests) or provide a validation of the level 3 cloud/IWC products (as the conclusion suggests). In the first case it should discuss its results in light of existing literature. In the second case it should compare values of cloud fraction and IWC with other datasets for validation. In the present state it reaches neither objective.

Minor Comments

Abstract : the abstract talks about "summer" several times, but it is unclear at this point that this means the NH summer. Please clarify the writing here (maybe by talking about months instead of seasons).

l.44: "the equatorial region of the NH": the equator is between the hemispheres, so this has no meaning. On the next line, "NH equator" has the same problem. Same thing on l.210 ("the equator of the NH"). Please find a correct way to say what you want to say (maybe reference latitudes).

l.54-57: the last two sentences of the abstract merely describe what was done in the paper, they do not convey what the work found out. Please remove them (see second main comment).

l.131-146: It is unclear why all the information provided here is relevant to the study. Please either connect these explanations to the results that are presented (for instance by arguing sampling limitations are connected somehow to the behaviour of the backscatter signal) or remove. The discussion of channels (l.132-135) is particularly confused: both 532 and 1064 backscatter coefficients are used to derive level 2 products. The 1064nm sentence has no verb.

l.154: please explain where does the IWC provided by the level 3 data come from? How what is retrieved and what are the uncertainties attached?

l.159: how is a "ice cloud-accepted sampled" defined? Please explain.

l.168: I understand excluding outside bins 1 and 44, but why exclude bins 17 and 18, which are near the center of the distribution? Please explain.

L.176: "altitude of 60m": do you mean a vertical resolution of 60m? Or it is 60m ASL?

l.184: Where do the maps shown in Fig. 1 and 2 come from? Did you create the data yourself? How did you do it? Did you derive them from the 3D gridded level 3 data? In fig. 2, only half of the colormap appears to be used, please use all the color range (e.g. set the max IWC at 0.005 g/m3).

l.194: see main comment about PSCs.

l.207: I do not understand what the authors mean when they say the ICF peaks under the "flat" tropopause altitude. ICF maximas are not at the tropopause altitude, they are well below. Do they mean that the highest ICF are found in the tropics? Why refer to the tropopause at all then? It is a known feature of the tropopause that it is constant within the tropical belt. Same comment for l.444.

l.207: please skip the definition of the tropopause

l.208: "... and decreases steadily towards the poles": the subject of "decreases" here is "the peak". Values of ICF do not decrease towards the poles, they even increase in some instances (e.g. Spring nighttime towards the South Pole). Maybe the authors meant that the altitude of ice clouds decreases. Please fix.

l.217-221: the limited sampling of nighttime data during the summer season in the NH polar region has a simple explanation: during JJA the NH polar regions is in mostly per-

Interactive
comment

manant daytime. So there is only very limited nighttime data. During DJF the NH polar region is in permanent nighttime, so there is only limited daytime data. The opposite is true for the SH polar region: permanent daytime in DJF, permanent nighttime in JJA. Some data is there, but not much. This explains the seasonal limited sampling of night-time and daytime data in polar regions. This is a fact related to the orbit of the Earth around the sun, that affects all observations, and not a CALIPSO limitation. Please clarify your discussion of this effect. The claim that Figure 3 "reveals" this well-known effect is a little exaggerated.

l.222:"extrapolation can be used for more complete and accurate data": Extrapolation basically fills out gaps in the data using existing information, but does not add information. Extrapolated data would not be more accurate.

l.230: here you attribute opposite (I think this is what you mean by "contradictory") variations of IWC and ICF to sampling biases. What do you mean by that? Sampling biases affect IWC and ICF detections in the same way: IWC cannot be retrieved where no cloud is detected. Please clarify.

l.234: The "spike-shaped structures" in Fig. 4 are a major concern. If the IWC data contains quality flags, the authors should see if raising the quality requirements make the spikes disappear. Otherwise, I would encourage the authors to contact the creators of the level 3 dataset and ask them about these spikes. These spikes do not look geophysical, and if they are recognized product artefacts efforts should be made to remove them from an article proposed for publication.

l.239: "we excluded the maximum..." The maximum was excluded from what? Please clarify.

l.263-265: this was already visible and clearer on Fig. 3. Please just reference the previous discussion.

l.265-267: Why is this interesting?

l.269: "negative values of the ICF": you mean a negative diurnal change of the ICF? Negative ICF should not exist.

Fig. 6: In Figure 6, rows 3 and 4 show basically the same thing: the number of points in which data has been sampled. A requirement for IWC retrieval is the detection of an ice cloud, so I am guessing that values shown in rows 3 and 4 should be the same or at least extremely close. Evaluating the difference between rows 3 and 4 would inform about how frequently an ice cloud is detected from which IWC cannot be retrieved, it would say more about the domain of validity of the IWC retrieval algorithm and less about the relationship between cloud presence and IWC. In Figure 6 differences between rows 3 and 4 cannot be seen anyway. Again, the sampling variability tells us more about the instrument than it teaches us about clouds. It is fine to discuss the instrument sampling if it allows a discussion about clouds afterwards, but by itself it is of limited interest. The limitations which are described here were already discussed elsewhere (see for instance the 2009 series of CALIPSO papers that discuss sampling in JAOT, e.g. Powell et al. 2009 and Hunt et al. 2009).

l.288: "we revealed some interesting facts..." All the facts explained below are already known. It would be more accurate to say that your results confirm known facts about how CALIPSO samples clouds.

l.291-293: this has already been discussed in sect. 3.3. Please sum up.

l.294: "this explains the behaviour..." Your statements do not explain the opposite trends, they are consistent with the opposite trends that were already discussed. Explaining the trends would mean 1) proposing a mechanism that could lead to opposite trends and 2) support the validity of that mechanism through literature or additional data. This has not been done here.

l.302-304: please compare and contrast your results with sub visual cirrus values from Martins et al. 2010 doi: 10.1029/2010JD014519 The total absence of SVC over convection centres is particularly surprising and should be discussed.

l.311: the values documented here might be correct, but why are they useful/important? Please explain.

l.326-328: this has already been described previously. Please sum up.

l.329-332: this has already been described for Fig. 6. Please avoid repetition.

l.335-368: all this is basically a verbal description of Fig. 8: this is smaller here, this is larger there. If these descriptions are not tied to an interpretation, that tries to make sense of the variations and explain how they are due to physical processes, they are basically useless. I might as well just look at the figure. These descriptions are a required step but are not sufficient. Please sum them up and point out to the reader which features are important and confirm or teach us things about ice clouds and IWC.

l.369-378: this part attempts to provide some explanations for the ice clouds and IWC features described by Fig. 8, but only considers possible instrument/sampling biases. As said before, discussions of instrument biases are interesting, but only if they allow you to ignore the biases and reveal accurate facts about geophysical quantities. The biases discussed here are already known.

Sect. 3.7: as stated in the main comments, I do not see the point of this section and the figures that go with it. Color scales of Fig. 9 mostly hide any possible correlations between the variables shown, but it appears the vast majority of RH and Temperatures are centered about a single main value, with little variability. The correlation coefficients suggest no correlation. Why the comparison should be done is not explained.

Sect.4: See last main comment. In its present state this section merely restates everything that has already been said before. It makes no attempt to explain why any of the results is important or useful or new. No context is provided, no literature cited. Please fix this.

l.472-476: Here the authors state that their analysis suggests that the distributions of ice cloud and IWC provided by the CALIPSO level 3 data are "reasonable and reliable".

There are several problems with this : 1) this goal was not presented as such in the abstract (i.e. the abstract does not say "in this paper we aim to show that the level 3 data are reasonable", it says "we aim to analyse the climatology of ice clouds and IWC"). 2) this goal has not been achieved: since you do not compare your level 3 statistics with literature or third-party data, there is no evidence in the article that suggests the values are reasonable and/or reliable. 3) the unexplained IWC spikes rather suggest that the level 3 IWC are in places neither reasonable or reliable. Product validation is an endeavour as important as studying cloud climatologies, but in this present state the paper has achieved neither of these goals.

Technical corrections

Please avoid "the CALIOP". Use "CALIOP" instead

l.42 : "On the other hand": please remove

l. 53: "(0.31)" this can be written as "(ICOD>0.3)".

L. 89-90: please put the citations in chronological order

l.110: "the active instruments like the CALIOP..." This sentence is not correct, please rewrite.

l.147: "the CALIPSO lidar instrument released": the lidar did not release a dataset. NASA did.

l.149: up to that point sentences were written in the present tense, now the writing switches to the past tense. Please fix the tenses.

l.165: the acronyms in equation 2 are not defined.

l.176: "as well as three files..." I don't understand. Please clarify.

l.227 and elsewhere: "latitude-altitude distributions" -> "zonal altitude distributions"

l.230: "and is attributed": what is the subject of that verb? Please fix the writing.

l.246: "to analyze... quantities": please remove.

l.257: "the bins above the Earth's surface": all the bins are above the Earth's surface. Please rephrase.

l.266: "nighttime which is below 5km": please fix the writing.

[Figure]

---

## Referee Comment (RC2) · Anonymous Referee #3 · 23 Apr 2020

Review of the Paper "A 10–year climatology of globally distributed ice cloud properties inferred from the CALIPSO observations" by H. Pan et al.

General comments This paper analyzes a 10 year climatology of the spatial and temporal distributions of ice cloud fraction, ice water content, and ice cloud optical depth for sub-visual, thin, and opaque clouds, based upon the newly released CALIPSO level 3 data files. Due to the concerns expressed in the Specific comments section of this review, the paper falls in the "Good" category. Revision is necessary prior to publication. Specific comments Though the calculations are useful, I do not know a) How the calculations compare to previous published calculations, and b) What is "new and innovative"

[Figure]

in the results presented in the paper. For these two reasons, the paper is problematic. The authors need to address these issues prior to acceptance of the paper. In the Introduction (line 92, page 4) the authors state that previous studies (6 studies, see lines 89-90) "are not sufficient". Why are those papers "not sufficient"? What does the current paper achieve that was not achieved by previous papers? Please answer these questions without stating an unsubstantiated negative value judgement. In the Summary and Conclusions section, there are no references to the previous literature. What are the commonalities (and differences) between the current calculations and the previous literature? Add a paragraph or two, with references, to address this concern in the Summary and Conclusions section. Technical comments Abstract, line 42: Change to "The latitude-and-altitude mean distributions of ICF and IWC were found to be unimodal in all seasons". I am not sure what unimodal refers to, either on line 42 or later in the text at lines 206 and 242. Please clarify. See comments below on lines 206-209. The use of the phrase "On the other hand" is confusing, since it is commonly used to make a contrast, and the sentence if it is used in (line 42) does not make a contrast to the previous sentence. Though the English in the text is generally good, there are several lines in the text which should be revised: Line 47, page 2: change "strong convective activities" to "strong convective activity" This comment also applies to lines 192 and 213. Line 111, page 5: change to the "A-train" constellation of satellites Line 131-136, page 6: change to The Level 1 CALIOP data file contains 532 nm parallel-polarized and 532 nm perpendicular-polarized attenuated backscatter coefficients. The attenuated backscatter coefficient at 1064 nm is also used to produce the level 2 data file products, given the CALIOP measurements and several algorithms.

In equation 2, line 165, on page 8: Why is the summation from 43 to 19 (with 43 below the $\sum$ symbol, instead of 19 to 43 with the 19 under the $\sum$ symbol?) Line 167, page 8 : What are the numerical ranges of IWCB and IWCH? Line 189, page 9 : change to storm activity Lines 206 − 209, page 10 : change to coverage of ICF generally exhibited a vertical profile with a single peak at ... independent tropical tropopause altitude, followed by a peak decreasing in altitude steadily towards both the SH and NH polar reg... "Why are the nighttime ICFl arger than the daytime values?" Though there is discussion later in the text (lines 248 and 373), it wou...

$change\,to\,asymmetrical\,distributions.\,For\,Figure\,1,\,page\,31,\,the\,color\,scale\,goes\,up\,to\,0.5,\,while\,the\,data\,is\,mainly\,from\,0\,to\,0.3.\,The$
$Use\,the\,same\,g/m3\,units\,in\,the\,text\,as\,in\,Figure\,5.\,Line\,235,\,page\,11$ :
$I\,did\,not\,understand\,what\,the\,``spike-shaped\,structure''\,refers\,to.\,From\,previous\,CALIPSO\,papers,\,this\,structure\,is\,likely\,iden$
$band\,lidar\,backscatter\,feature,\,or\,something\,else?)\,Line\,238\quad-\quad240,\,page\,11$ :
$The\,sentence\,is\,not\,clear.\,Please\,revise.\,The\,term\,``we\,excluded\,the\,maximum''\,is\,not\,clear.\,Line\,328,\,page\,15$ :
$change\,to\,during\,night\,compared\,to\,day.\,In\,Figure\,8\,(page\,37),\,it\,may\,be\,better\,to\,graph\,100\,(\#night\,observations$
$-\#day\,observations)/\#night\,observations\,instead\,of\,\#night\,observations\quad-$
$\#day\,observations.\,Line\,387,\,page\,18:The\,phrase\,``the\,values\,of\,these\,parameters\,were\,obtained\,from\,the\,CALIPSO\,platform''$
$389,\,page\,18:change\,to\,Fig.\,9\,shows\,the\,10-year\,global\,contour\,density\,plots\,of\,nighttime\,and\,daytime\,IWC,\,RH,\,and\,TE.\,Line\,44$
$446,\,page\,20:Rephrase.\,See\,comment\,on\,lines\,206-209.\,Table\,3.\,The\,numbers\,are\,too\,small.\,Expand\,the\,table\,into\,two\,parts\,to\,incr$

1. Does the paper address relevant scientific questions within the scope of ACP? yes
2. Does the paper present novel concepts, ideas, tools, or data? No (the calculations are similar to those in many previous studies) 3. Are substantial conclusions reached? No 4. Are the scientific methods and assumptions valid and clearly outlined? Yes 5. Are the results sufficient to support the interpretations and conclusions? There are few interpretations and few conclusions that are new. 6. Is the description of experiments and calculations sufficiently complete and precise to allow their reproduction by fellow scientists (traceability of results)? Yes 7. Do the authors give proper credit to related work and clearly indicate their own new/original contribution? No (this is a major concern) 8. Does the title clearly reflect the contents of the paper? Yes 9. Does the abstract provide a concise and complete summary? Yes 10. Is the overall presentation well structured and clear? Yes 11. Is the language fluent and precise? There are several places in the text where the English could be improved. 12. Are mathematical formulae, symbols, abbreviations, and units correctly defined and used? Mostly yes (the bounds of the sigma summation are odd) 13. Should any parts of the paper (text, formulae, figures, tables) be clarified, reduced, combined, or eliminated? There are several places where clarifications are needed, and the fonts of Table 3 are too small. 14. Are the number and quality of references appropriate? Suggested additional paragraphs will introduce more references. Is the amount and quality of

supplementary material appropriate? Not relevant here.

Please also note the supplement to this comment:
https://www.atmos-chem-phys-discuss.net/acp-2019-1116/acp-2019-1116-RC2-supplement.pdf

[Figure]

**Supplement:**

Review of the Paper "A 10–year climatology of globally distributed ice cloud properties inferred from the CALIPSO observations" by H. Pan et al.

General comments

This paper analyzes a 10 year climatology of the spatial and temporal distributions of ice cloud fraction, ice water content, and ice cloud optical depth for sub-visual, thin, and opaque clouds, based upon the newly released CALIPSO level 3 data files.

Due to the concerns expressed in the Specific comments section of this review, the paper falls in the "Good" category. Revision is necessary prior to publication.

Specific comments

Though the calculations are useful, I do not know

a) How the calculations compare to previous published calculations, and
b) What is "new and innovative" in the results presented in the paper.

For these two reasons, the paper is problematic. The authors need to address these issues prior to acceptance of the paper.

In the Introduction (line 92, page 4) the authors state that previous studies (6 studies, see lines 89-90) "are not sufficient". Why are those papers "not sufficient"? What does the current paper achieve that was not achieved by previous papers? Please answer these questions without stating an unsubstantiated negative value judgement.

In the Summary and Conclusions section, there are no references to the previous literature. What are the commonalities (and differences) between the current calculations and the previous literature? Add a paragraph or two, with references, to address this concern in the Summary and Conclusions section.

Technical comments

Abstract, line 42: Change to "The latitude-and-altitude mean distributions of ICF and IWC were found to be unimodal in all seasons". I am not sure what unimodal refers to, either on line 42 or later in the text at lines 206 and 242. Please clarify. See comments below on lines 206-209.

The use of the phrase "On the other hand" is confusing, since it is commonly used to make a contrast, and the sentence if it is used in (line 42) does not make a contrast to the previous sentence.

Though the English in the text is generally good, there are several lines in the text which should be revised:

Line 47, page 2: change "strong convective activities" to "strong convective activity" This comment also applies to lines 192 and 213.

Line 111, page 5: change to      the "A-train" constellation of satellites

Line 131-136, page 6: change to    The Level 1 CALIOP data file contains 532 nm parallel-polarized and 532 nm perpendicular-polarized attenuated backscatter coefficients. The attenuated

backscatter coefficient at 1064 nm is also used to produce the level 2 data file products, given the CALIOP measurements and several algorithms.

In equation 2, line 165, on page 8: Why is the summation from 43 to 19 (with 43 below the $\sum$ symbol, instead of 19 to 43 with the 19 under the $\sum$ symbol?)

Line 167, page 8: What are the numerical ranges of IWCBB and IWCH?

Line 189, page 9: change to     storm activity

Lines 206-209, page 10: change to       coverage of ICF generally exhibited a vertical profile with a single peak, with the peak under the latitude-independent tropical tropopause altitude, followed by a peak decreasing in altitude steadily towards both the SH and NH polar regions.

When I first looked at Fig. 3, I asked myself the question: "Why are the nighttime ICF larger than the daytime values?" Though there is discussion later in the text (lines 248 and 373), it would be good to tell the reader that a discussion of this matter is discussed later in the text. Is the nighttime and daytime differences a measurement artifact or a cloud microphysics issue? Are nighttime temperatures less than daytime temperatures? Some reference to the previous literature would be helpful.

Line 228, Page 11: change to     asymmetrical distributions.

For Figure 1, page 31, the color scale goes up to 0.5, while the data is mainly from 0 to 0.3. The authors should consider redoing the graph with a color scale from 0 to 0.3

For Figure 2, page 31, the color scale goes up to 0.01, while the data is mainly from 0 to 0005. The authors should consider redoing the graph with a color scale from 0 to 0.005

Line 232, page 11: Use the same $g/m^3$ units in the text as in Figure 5.

Line 235, page 11: I did not understand what the "spike-shaped structure" refers to. From previous CALIPSO papers, this structure is likely identified with a physical feature. Refer to the literature to make the structure less mysterious. (Is it related to the melting-band lidar backscatter feature, or something else?)

Line 238-240, page 11: The sentence is not clear. Please revise. The term "we excluded the maximum" is not clear.

Line 328, page 15: change to     during night compared to day.

In Figure 8 (page 37), it may be better to graph

     100 (# night observations – # day observations) / # night observations

instead of # night observations - # day observations.

Line 387, page 18: The phrase "the values of these parameters were obtained from the CALIPSO platform" implies that the RH and temperature profiles are measured by the CALIPSO experiment. It would be better to state that auxiliary files specify these profiles. Please specify the origin of the auxiliary files.

Line 388-389, page 18: change to    Fig. 9 shows the 10-year global contour density plots of nighttime and daytime IWC, RH, and TE.

Line 443-446, page 20: Rephrase. See comment on lines 206-209.

Table 3. The numbers are too small. Expand the table into two parts to increase the font size.

1. Does the paper address relevant scientific questions within the scope of ACP? yes

2. Does the paper present novel concepts, ideas, tools, or data? No (the calculations are similar to those in many previous studies)

3. Are substantial conclusions reached? No

4. Are the scientific methods and assumptions valid and clearly outlined? Yes

5. Are the results sufficient to support the interpretations and conclusions? There are few interpretations and few conclusions that are new.

6. Is the description of experiments and calculations sufficiently complete and precise to allow their reproduction by fellow scientists (traceability of results)? Yes

7. Do the authors give proper credit to related work and clearly indicate their own new/original contribution? No (this is a major concern)

8. Does the title clearly reflect the contents of the paper? Yes

9. Does the abstract provide a concise and complete summary? Yes

10. Is the overall presentation well structured and clear? Yes

11. Is the language fluent and precise? There are several places in the text where the English could be improved.

12. Are mathematical formulae, symbols, abbreviations, and units correctly defined and used? Mostly yes (the bounds of the sigma summation are odd)

13. Should any parts of the paper (text, formulae, figures, tables) be clarified, reduced, combined, or eliminated? There are several places where clarifications are needed, and the fonts of Table 3 are too small.

14. Are the number and quality of references appropriate? Suggested additional paragraphs will introduce more references.

15. Is the amount and quality of supplementary material appropriate? Not relevant here.

---

## Author Comment (AC1) · 10 Jun 2020

Response to reviewers comments

Article Reference: acp-2019-1116

Title: "Comparison of CALIPSO level 3 cloud products and application of diurnal vertical variation of global ice cloud properties" by Honglin Pan et al. Journal: Atmospheric Chemistry and Physics (ACP)

Dear Editor, Thank you very much for reviewing our research paper and providing the list of comments/suggestions raised by the learned reviewers which in turn helped us

in improving the quality of an earlier version of the paper. As per the suggestions of reviewers, we have gone through the entire paper giving suitable replies to their queries and revised the whole paper to correct presentation errors, as well as add some figures to the supplementary material. We deleted some of the contents of manuscript which are repetitions following the suggestions given by the reviewers.

The authors wish to thank the Handling Editor for his encouragement and support in contacting the reviewers to complete the peer-review process in time at the earliest. The authors are also grateful to the two anonymous reviewers for their constructive and useful comments which in turn improved the scientific content of an earlier version of the manuscript.

The modified text in the revised manuscript is highlighted with yellow color to distinguish with the normal text. However, the tracked version of submitted manuscript provided the changes/corrections implemented following the reviewer(s) suggestions is attached during the revision submission process.

Anonymous Reviewer # 1 Reply to major comments: First of all, we really appreciate your instructive and scientific suggestions on our paper. And we have revised them thoroughly following your suggestions and replies are given below. For the first point, we agree the idea that average ice cloud detections over entire vertical profiles (between -0.5 km and 20 km ASL) to build maps of ice cloud fraction is not reasonable. Therefore, we group the ICF into three types, that is, low, middle, and high to recalculate the result of ICF over the globe based on the cloud top pressure, according to Chepfer et al., 2010. doi: 10.1029/2009JD012251 that you listed, Thanks again! For the second suggestion, we are not agreed the suggestion. Namely, to eliminate PSCs from the dataset before creating any statistics. The reasons are as follows: The CALIPSO level 3 ice product is aggerated from newest version level 2 cloud product, and ice cloud samples are selected by a series of algorithms and quality assurance. Namely, the cloud aerosol discrimination (CAD algorithm) is used to distinguish the cloud and aerosol by the score of -100 to +100, and when the score bigger than 20,

the feature is identified to cloud, otherwise to aerosol. Moreover, the aerosol also can be further grouped into tropospheric aerosol and stratospheric aerosol, the latter includes PSC aerosol, volcanic ash, and sulfate/other. Then, the cloud phase algorithm can classify the cloud into water and ice phase. Consequently, ICF retrieved from ice cloud product should be eliminating the PSC aerosol to the large extent, and the PSC is included into the ice cloud, which would be reasonable. In addition, CALIPSO also released level 3 stratospheric aerosol product in 2018, which can further believe the ice cloud product does not contain the PSC aerosol to some extent. And the detailed quality assurance of ice cloud product has been added into the manuscript. For the third point, we accept the suggestion you provided. I drop the section of relationship between ice cloud properties and meteorological variable of relative humidity and temperature. For the fourth point, thank you for your suggestion very much. In order to more concrete study objective of our paper, we changed the title to "Comparison of CALIPSO level 3 cloud product and application to diurnal vertical variation of global ice cloud properties". That is, the detailed study content is also corresponding to change that we first compare the ICF from CALIPSO with MODIS, and then we used the CALIPSO ice cloud product to study the vertical distribution of diurnal variability of ice cloud properties. Accordingly, the present version of manuscript has changed a lot than the previous version. In the end, we have discussed the previous study result to our results in the last section. At present, the new manuscript is more paid attention to the vertical variation of ice cloud characteristics. However, some caveats exist when using CALIPSO level 3 version 1.0 profile product in the study. For example, polar orbiting satellite data with a repeat cycle of 16 days and a local equator-crossing times (EXTs) of 1:30 a.m./p.m. which is limited to conduct analysis of diurnal variability of ice cloud properties, and sun light noise caused the bad data quality in the daytime. Therefore, in this paper, we cannot analyze the real physical process of diurnal variation of ice clouds due to some artificial variation, but we can describe the observational phenomenon of ice cloud, which can pave the way for further study of ice clouds, such as cloud model. Further, as current climate models can not accurately depict the vertical distribution of ice clouds, especially on the diurnal difference on the vertical resolving scale, CALIOP do provide a valuable opportunity to study the diurnal variations of global ice cloud properties. In summary, we think our results are favorable to improve the modeling of ice clouds as the observational facts. The replies we have provided are best up to our knowledge and hope the replies are given satisfactorily. Replies to the minor comments: Abstract: the abstract talks about "summer" several times, but it is unclear at this point that this means the NH summer. Please clarify the writing here (may be by talking about months instead of seasons). Answer: Thank you for your suggestion. We have corrected it. l.44: "the equatorial region of the NH": the equator is between the hemispheres, so this has no meaning. On the next line, "NH equator" has the same problem. Same thing on l.210 ("the equator of the NH"). Please find a correct way to say what you want to say (maybe reference latitudes). Answer: Thank you for your suggestion. We have corrected it. l.54-57: the last two sentences of the abstract merely describe what was done in the paper, they do not convey what the work found out. Please remove them (see second main comment). Answer: Thank you for your suggestion. We have removed it. l.131-146: It is unclear why all the information provided here is relevant to the study. Please either connect these explanations to the results that are presented (for instance by arguing sampling limitations are connected somehow to the behavior of the backscatter signal) or remove. The discussion of channels (l.132-135) is particularly confused: both 532 and 1064 backscatter coefficients are used to derive level 2 products. The 1064nm sentence has no verb. Answer:Thank you for your suggestion. We have removed it. l.154: please explain where does the IWC provided by the level 3 data come from? How what is retrieved and what are the uncertainties attached? Answer:The level 3 data of IWC come from level 2 cloud product retrieved from CALIPSO. The retrieved formula is from where $\sigma$ is 532nm the volume extinction coefficient of km-1, and C0 = 119 g.cm-1 and C1 = 1.22 g.cm-1 are parameters obtained from an observed empirical relationship between lidar extinction and in situ measurements of cloud properties (Zhao et al., 2018; Pan et al., 2017). Consequently, the uncertainties due to the empirical relationship of the formula.We have added and explained this content in this paper. l.159: how is a "ice cloud-accepted sampled" defined? Please explain. Answer:Ice cloud-accepted samples refer to the number of ice clouds samples passed the quality filters including quality assurance and quality control, respectively. And the detailed content can be found in the section 2.3 of the paper. l.168: I understand excluding outside bins 1 and 44, but why exclude bins 17 and 18, which are near the center of the distribution? Please explain. Answer: Bin 1 and 44 are very small or big, and small magnitude values with less confidence are stored in bin 17 and 18 from Table 1 below. So, we exclude them to make the result more reasonable.

l.176: "altitude of 60m": do you mean a vertical resolution of 60m? Or it is 60m ASL? Answer: Yes, it is 60m ASL. l.184: Where do the maps shown in Fig. 1 and 2 come from? Did you create the data yourself? How did you do it? Did you derive them from the 3D gridded level 3 data? In fig. 2, only half of the colormap appears to be used, please use all the color range (e.g. set the max IWC at 0.005 g/m3). Answer: Yes. Figure 1 and 2 come from the 3D gridded level 3 data. We make an integral of altitude samples at each latitude and longitude bin for the CALIOP L3 Ice Cloud product, transforming ICF from 3D into 2D.And we deleted the Fig.1 and 2 because we changed the study content and title, now paid mainly attention to vertical distribution of diurnal variation of ice cloud properties. l.194: see main comment about PSCs. Answer: We reckon that no need to eliminate PSCs from the dataset before creating any statistics. The reasons are as follows: The CALIPSO level 3 ice products is aggerated from newest version level 2 cloud products, and ice cloud samples are selected by a series of algorithms and quality assurance. Namely, the cloud aerosol discrimination (CAD algorithm) is used to distinguish the cloud and aerosol by the score of -100 to +100, and when the score bigger than 20, the feature is identified to cloud, otherwise to aerosol. Moreover, the aerosol also can be further grouped into tropospheric aerosol and stratospheric aerosol, the latter includes PSC aerosol, volcanic ash, and sulfate/other. Then, the cloud phase algorithm can classify the cloud into water and ice phase. Consequently, ICF retrieved from ice cloud product should be eliminating

the PSC aerosol to the large extent, and the PSC is included into the ice cloud, which would be reasonable. In addition, CALIPSO also released level 3 stratospheric aerosol product in 2018, which can further believe the ice cloud product does not contain the PSC aerosol to some extent. And the detailed quality assurance of ice cloud product has been added into the manuscript. l.207: I do not understand what the authors mean when they say the ICF peaks under the "flat" tropopause altitude. ICF maximas are not at the tropopause altitude, they are well below. Do they mean that the highest ICF are found in the tropics? Why refer to the tropopause at all then? It is a known feature of the tropopause that it is constant within the tropical belt. Same comment for l.444. Answer: Yes, the highest ICF are found in the tropics. We want to describe that the coverage of ICF generally exhibited a flattened cosine curve distribution, caused the strong mixing and heat transport by the Hadley circulations, where the peak is below the tropical tropopause. In addition, to clearly state the phenomenon, we have added the tropopause displayed in black solid line on the ICF of zonal altitude plot. l.207: please skip the definition of the tropopause Answer: Thank you for your suggestion. We have skipped it. l.208: "... and decreases steadily towards the poles": the subject of "decreases" here is "the peak". Values of ICF do not decrease towards the poles, they even increase in some instances (e.g. Spring nighttime towards the South Pole). Maybe the authors meant that the altitude of ice clouds decreases. Please fix. Answer: Thank you for your suggestion. We have fixed it. l.217-221: the limited sampling of nighttime data during the summer season in the NH polarregionhasasimpleexplanation: duringJJAtheNHpolarregionsisinmostlypermanent daytime. So, there is only very limited nighttime data. During DJF the NH polar region is in permanent nighttime, so there is only limited daytime data. The opposite is true for the SH polar region: permanent daytime in DJF, permanent nighttime in JJA. Some data is there, but not much. This explains the season all limited sampling of nighttime and daytime data in Polar Regions. This is a fact related to the orbit of the Earth around the sun, which affects all observations, and not a CALIPSO limitation. Please clarify your discussion of this effect. The claim that Figure 3 "reveals" this well-known effect is a little exaggerated.

Answer: Thank you for your suggestion. We have added the reason you listed into our paper. l.222:"extrapolation can be used for more complete and accurate data": Extrapolation basically fills out gaps in the data using existing information, but does not add information. Extrapolated data would not be more accurate. Answer: Thank you for your suggestion. We have removed it. l.230: here you attribute opposite (I think this is what you mean by "contradictory") variations of IWC and ICF to sampling biases. What do you mean by that? Sampling biases affect IWC and ICF detections in the same way: IWC cannot be retrieved where no cloud is detected. Please clarify. Answer: Thank you for your suggestion.We calculated the IWC by the sample sum of the bin in 2-16 and bin in 19-43. And we found in Fig.6 that more (less) samples of IWC in large bin of 19-43 (small bin of 2-16) in the daytime than the nighttime, which lead to the IWC in the daytime bigger than the nighttime. And we have re-write the section in the paper.

l.234: The "spike-shaped structures" in Fig. 4 are a major concern. If the IWC data contains quality flags, the authors should see if raising the quality requirements makes thespikesdisappear. Otherwise, I would encourage the authors to contact the creators of the level 3 dataset and ask them about these spikes. These spikes do not look geophysical, and if they are recognized product artefacts efforts should be made to remove them from an article proposed for publication. Answer: Thank you for your suggestion. We used the quality flags to re-plot the results and adjust the color range. We find the results still occur the "spike-shaped structures" but the spikesbecame weaker. So, we also think the spikes possibly are the artifacts based on the retrieved algorithm of IWC. And we have emailed the contact the creator of this product, but without the response. Now, we only point out the spikes are artifact in the paper. l.239: "we excluded the maximum..." The maximum was excluded from what? Please clarify. Answer: We excluded the maximum of the IWC (in Fig.5 that we re-plot, the small red rectangle contains one IWC data over the SH polar region in summer, the big red rectangle zooms in on the small red rectangle) because the value lager than 0.01g/m3. l.263-265: this was already visible and clearer on Fig. 3. Please just reference the previous discussion. Answer: Thank you for your suggestion. We have corrected it. l.265-267: Why

is this interesting? Answer: Because the total number of samples for the daytime was smaller than nighttime in most of levels. However, the total number of samples for the daytime was more than nighttime which is below ~5 km over the low and middle latitude for both hemispheres (because the x-axis is on the logarithmic scale and hence, the negative values were neglected). l.269: "negative values of the ICF": you mean a negative diurnal change of the ICF? Negative ICF should not exist. Answer: Yes, a negative diurnal change of the ICF. Fig. 6: In Figure 6, rows 3 and 4 show basically the same thing: the number of points in which data has been sampled. A requirement for IWC retrieval is the detection of an ice cloud, so I am guessing that values shown in rows 3 and 4 should be the same or at least extremely close. Evaluating the difference between rows 3 and 4 would inform about how frequently an ice cloud is detected from which IWC cannot be retrieved, it would say more about the domain of validity of the IWC retrieval algorithm and less about the relationship between cloud presence and IWC. In Figure 6 differences between rows 3 and 4 cannot be seen anyway. Again, the sampling variability tells us more about the instrument than it teaches us about clouds. It is fine to discuss the instrument sampling if it allows a discussion about clouds afterwards, but by itself it is of limited interest. The limitations which are described here were already discussed elsewhere (see for instance the 2009 series of CALIPSO papers that discuss sampling in JAOT, e.g. Powell et al. 2009 and Hunt et al. 2009). AnswerïijŽThank you for the reference you listed, and we are agreed to your idea about the point. However, in this paper, we want to emphasize the total number of IWC and ICF samples show the similar distribution during nighttime and daytime, which can be further verify the negative diurnal variation of IWC owing to more lager values in the day are sampled (discussed in Fig.6). Additionally, we re-write this section in the manuscript. l.288: "we revealed some interesting facts..." All the facts explained below are already known. It would be more accurate to say that your results confirm known facts about how CALIPSO samples clouds. Answer: Thank you for your suggestion. We have corrected it. l.291-293: this has already been discussed in sect. 3.3. Please sum up. Answer: Thank you for your suggestion. We have summed them up. l.294:

"this explains the behavior..." Your statements do not explain the opposite trends, they are consistent with the opposite trends that were already discussed. Explaining the trends would mean 1) proposing a mechanism that could lead to opposite trends and 2) support the validity of that mechanism through literature or additional data. This has not been done here. Answer: Thank you for your suggestion. We have explained it in the paper. We calculated the IWC by the sample sum of the bin in 2-16 and bin in 19-43 (see the Table 1). And we found in Fig.6 that more (less) samples of IWC in large bin of 19-43 (small bin of 2-16) in the daytime than the nighttime, which lead to the IWC in the daytime bigger than the nighttime. And we found the total number of IWC and ICF samples show the similar distribution during nighttime and daytime, which can be further verify the negative diurnal variation of IWC owing to more lager values in the day are sampled, which present the opposite trends of ICF.

l.302-304: please compare and contrast your results with sub visual cirrus values from Martins et al. 2010 doi: 10.1029/2010JD014519 the total absence of SVC over convection centers is particularly surprising and should be discussed. Answer: Thank you for your suggestion. We have compared and discussed it in our paper. And the results of Fig.3 and 5 from Martins about the sub-visual ice cloud are basically similar to our results of Fig.8. In addition, the paper is published in 2011 rather than 2010. l.311: thevaluesdocumentedheremightbecorrect,butwhyaretheyuseful/important? Please explain. Answer: We have dropped this part.In order to more concrete study objective of our paper, we changed the title to "Comparison of CALIPSO level 3 cloud product and application to diurnal vertical variation of global ice cloud properties". That is, the detailed study content is also corresponding to change that we first compare the ICF from CALIPSO with MODIS, then we used the CALIPSO ice cloud product to study the vertical distribution of diurnal variability of ice cloud properties. Accordingly, the present version of manuscript has changed a lot than the previous version. At present, the new manuscript is more paid attention on the vertical variation of ice cloud characteristics. l.326-328: this has already been described previously. Please sum up. Answer: Thank you for your suggestion. We have summed them up. l.329-332: this

has already been described for Fig. 6. Please avoid repetition. Answer: Thank you for your suggestion. We have deleted them. l.335-368: all this is basically a verbal description of Fig. 8: this is smaller here; this is larger there. If these descriptions are not tied to an interpretation, that tries to make sense of the variations and explain how they are due to physical processes, they are basically useless. I might as well just look at the figure. These descriptions are a required step but are not sufficient. Please sum them up and point out to the reader which features are important and confirm or teach us things about ice clouds and IWC. Answer: Thank you for your suggestion. We have summed them up. l.369-378: this part attempts to provide some explanations for the ice clouds and IWC features described by Fig. 8, but only considers possible instrument/sampling biases. As said before, discussions of instrument biases are interesting, but only if they allow you to ignore the biases and reveal accurate facts about geophysical quantities. The biases discussed here are already known. Answer: The diurnal variation mainly contains artificial variation (e.g. instrument-induced, classification-induced, and sampling-induced variabilities) and real variation of physical process of ice clouds that you said, but the observations of ice clouds based on CALIPSO retrieves are not reasonable if we ignore the instrument biases. So, we want to express analyzing the diurnal variation of ice clouds should considered these factors by this part. Moreover, as current climate models can not accurately depict the vertical distribution of ice clouds, especially on the diurnal difference on the vertical resolving scale, CALIOP do provide a valuable opportunity to study the diurnal variations of global ice cloud properties. So, we think our findings of the variation of ice properties in vertically resolved scale by CALIPSO level 3 global monthly ice cloud data, are favorable to improve the modeling of ice clouds. Sect. 3.7: as stated in the main comments, I do not see the point of this section and the figures that go with it. Color scales of Fig. 9 mostly hide any possible correlations between the variables shown, but it appears the vast majority of RH and Temperatures are centered about a single main value, with little variability. The correlation coefficients suggest no correlation. Why the comparison should be done is not explained. Answer: We have deleted this

Interactive
comment

section based on your suggestion based on your major comment. Sect.4: See last main comment. In its present state this section merely restates everything that has already been said before. It makes no attempt to explain why any of the results is important or useful or new. No context is provided, no literature cited. Please fix this. Answer: Thank you for your suggestion. We have fixed them in the last section that cited the literature from previous studies. l.472-476: Here the authors state that their analysis suggests that the distributions of icecloudandIWCprovidedbytheCALIP-SOlevel3dataare"reasonableandreliable".There are several problems with this : 1) this goal was not presented as such in the abstract (i.e. the abstract does not say "in this paper we aim to show that the level 3 data are reasonable", it says "we aim to analyze the climatology of ice clouds and IWC"). 2) this goal has not been achieved: since you do not compare your level 3 statistics with literature or third-party data, there is no evidence in the article that suggests the values are reasonable and/or reliable. 3) the unexplained IWC spikes rather suggest that the level 3 IWC are in places neither reasonable or reliable. Product validation is an endeavor as important as studying cloud climatology, but in this present state the paper has achieved neither of these goals. Answer: Thank you for your suggestion. We have compared level 3 ice cloud product from CALIPSO with MODIS level 3 cloud products, and we also compare our results with the previous studies. Answer to technical corrections Please avoid "the CALIOP". Use "CALIOP" instead Answer: Thank you for your suggestion. We have corrected it. l.42: "On the other hand": please remove Answer: Thank you for your suggestion. We have removed it. l. 53: "(0.31)" this can be written as "(ICOD>0.3)". Answer: Thank you for your suggestion. We have re-written it. l. 89-90: please put the citations in chronological order Answer: Thank you for your suggestion. We have corrected it. l.110: "the active instruments like the CALIOP..." This sentence is not correct, please rewrite. Answer: Thank you for your suggestion. We have re-written it. l.147: "the CALIPSO lidar instrument released": the lidar did not release a dataset. NASA did. Answer: Thank you for your suggestion. We have corrected it. l.149: up to that point sentences were written in the present tense, now the writing switches to the past tense.

Please fix the tenses. Answer: Thank you for your suggestion. We have re-written it. l.165: the acronyms in equation 2 are not defined. Answer: Thank you for your suggestion. We have added it. l.176: "as well as three files..." I don't understand. Please clarify. Answer: Thank you for your suggestion. We have re-written it. l.227 and elsewhere: "latitude-altitude distributions" -> "zonal altitude distributions" Answer: Thank you for your suggestion. We have corrected it. l.230: "and is attributed": what is the subject of that verb? Please fix the writing. Answer: Thank you for your suggestion. We have re-written it. l.246: "to analyze... quantities": please remove. Answer: Thank you for your suggestion. We have removed it. l.257: "the bins above the Earth's surface": all the bins are above the Earth's surface. Please rephrase. Answer: Thank you for your suggestion. We have re-written it. l.266: "nighttime which is below 5km": please fix the writing. Answer: Thank you for your suggestion. We have corrected it.

---

## Author Comment (AC2) · 10 Jun 2020

Response to reviewers comments

Article Reference: acp-2019-1116

Title: "Comparison of CALIPSO level 3 cloud products and application of diurnal vertical variation of global ice cloud properties" by Honglin Pan et al. Journal: Atmospheric Chemistry and Physics (ACP)

Anonymous Reviewer # 2 Answer to major comments: First of all, we really appreciate your instructive and scientific suggestions to our paper! And we have revised them

thoroughly below. Answer to Specific comments Though the calculations are useful, I do not know a) How the calculations compare to previous published calculations, and b) What is "new and innovative" in the results presented in the paper. For these two reasons, the paper is problematic. The authors need to address these issues prior to acceptance of the paper. Answer: Thank you for your suggestion very much. In order to more concrete study objective of our paper, we changed the title to "Comparison of CALIPSO level 3 cloud product and application to diurnal vertical variation of global ice cloud properties". That is, the detailed study content is also corresponding to change that we first compare the ICF from CALIPSO with MODIS, and then we used the CALIPSO ice cloud product to study the vertical distribution of diurnal variability of ice cloud properties. Accordingly, the present version of manuscript has changed a lot than the previous version. In the end, we have discussed the previous study result to our results in the last section. At present, the new manuscript is more paid attention to the vertical variation of ice cloud characteristics. However, some caveats exist when using CALIPSO level 3 version 1.0 profile product in the study. For example, polar orbiting satellite data with a repeat cycle of 16 days and a local equator-crossing times (EXTs) of 1:30 a.m./p.m. which is limited to conduct analysis of diurnal variability of ice cloud properties, and sun light noise caused the bad data quality in the daytime. Therefore, in this paper, we cannot analyze the real physical process of diurnal variation of ice clouds due to some artificial variation, but we can describe the observational phenomenon of ice cloud, which can pave the way for further study of ice clouds, such as cloud model. Further, as current climate models can not accurately depict the vertical distribution of ice clouds, especially on the diurnal difference on the vertical resolving scale, CALIOP do provide a valuable opportunity to study the diurnal variations of global ice cloud properties. In summary, we think our results are favorable to improve the modeling of ice clouds as the observational facts. In the Introduction (line 92, page 4) the authors state that previous studies (6 studies, see lines 89-90) "are not sufficient". Why are those papers "not sufficient"? What does the current paper achieve that was not achieved by previous papers? Please

answer these questions without stating an unsubstantiated negative value judgement. Answer: Because many of these studies have been limited to small spatial regions and short temporal periods, as well as certain classifications of ice clouds. Therefore, long-term and large-scale studies of ice clouds resolved in the vertical are necessary, to delineate some specific physical processes, which can be regarded as the cause of the biggest errors in cloud modeling. In the Summary and Conclusions section, there are no references to the previous literature. What are the commonalities (and differences) between the current calculations and the previous literature? Add a paragraph or two, with references, to address this concern in the Summary and Conclusions section. Answer: Thank you for your suggestion. We have added a paragraph to compare the previous study with our results in the Summary and Conclusions section. The detailed contents are as follows: Our results can be compared with early studies. Huang et al. (2015) used the CALIPSO and CloudSat cloud data to investigate the climatology of cloud water content with different cloud type. Figure 2 of their study reveal that each type of cloud shows an asymmetric distribution between NH and SH. And over both middle and high latitudes, clouds present lager seasonal variation in the NH than SH. The vertical distributions of ICF and IWC in Fig.4 and 5 show the similar characteristics. However, we also find that the PSCs occur in the SH. Figure A3 of Sassen et al. (2008) illustrated that maximum difference of cirrus occurrence over the tropics during diurnal cycle. This point of their results is confirmed by our analysis of Fig.7. Hong and Liu. (2015) suggested that thin ice clouds frequently occur in the tropics at high altitudes, and infrequent occurrence of sub-visual ice clouds. Our study of Fig.8 indicates that sub-visual ice clouds are more frequent at high altitudes (above 15km) than low altitudes, and thin ice clouds frequent occurrence in the low and middle latitudes at high altitudes. In addition, Fig.5 of Martins et al. (2011) revealed the sub-visual ice cloud showed the maximum occurrence frequency at ∼15 km over the tropics, but at ∼10km over mid latitudes. Our results of Fig.8 demonstrated that sub-visual ice clouds with the maximum concurrency at ∼15km over the mid-latitudes by CALIPSO data. Moreover, their results of Fig.3 showed the sub-visual ice clouds

with about 30%-40% of ICF in the strong convective areas, which are consisted with our results of Fig.2. Answer to technical comments Abstract, line 42: Change to "The latitude-and-altitude mean distributions of ICF and IWC were found to be unimodal in all seasons". I am not sure what unimodal refers to, either on line 42 or later in the text at lines 206 and 242. Please clarify. See comments below on lines 206-209. Answer: Thank you for your suggestion. We have re-written them. The use of the phrase "On the other hand" is confusing, since it is commonly used to make a contrast, and the sentence if it is used in (line 42) does not make a contrast to the previous sentence. Answer: Thank you for your suggestion. We have dropped them. Though the English in the text is generally good, there are several lines in the text which should be revised: Line 47, page 2: change "strong convective activities" to "strong convective activity" This comment also applies to lines 192 and 213. Answer: Thank you for your suggestion. We have corrected them. Line 111, page 5: change to the "A-train" constellation of satellites Answer: Thank you for your suggestion. We have re-written them. Line 131-136, page 6: change to The Level 1 CALIOP data file contains 532 nm parallel polarized and 532 nm perpendicular-polarized attenuated backscatter coefficients. The attenuated backscatter coefficient at 1064 nm is also used to produce the level 2 data file products, given the CALIOP measurements and several algorithms. Answer: Thank you for your suggestion. But we have dropped them, because we revised content of whole paper to better make the study objective more concrete. In equation 2, line 165, on page 8: Why is the summation from 43 to 19 (with

43 below the $\sum symbol, instead of 19 to 43 with the 19 under the \sum symbol?) Answer$ :
$Thank you for your suggestion. We made a mistake about it, and we have re-$
$written the equation from bin 19 to bin 43. Line 167, page 8$ :
$What are the numerical ranges of IWCB B and IWCH? Answer$ :
$The histogram bin range of IWC can be seen in Table 1 below. We only use the bin 2-$
$16 and bin 19 - 43 to calculate IWC, and the bin 1 and 44, as well as 17 and 18 are neglected.$

Line 189, page 9: change to storm activity Answer: Thank you for your suggestion. We have re-written them. Lines 206-209, page 10: change to coverage of ICF generally exhibited a vertical profile with a single peak, with the peak under the latitude-independent tropical tropopause altitude, followed by a peak decreasing in altitude steadily towards both the SH and NH polar regions. Answer: Thank you for your suggestion. We have re-written them. When I first looked at Fig. 3, I asked myself the question: "Why are the nighttime ICF larger than the daytime values?" Though there is discussion later in the text (lines 248 and 373), it would be good to tell the reader that a discussion of this matter is discussed later in the text. Are the nighttime and daytime differences a measurement artifact or a cloud microphysics issue? Are nighttime temperatures less than daytime temperatures? Some reference to the previous literature would be helpful. Answer: Thank you for your suggestion. We have added them in the manuscript. Line 228, Page 11: change to asymmetrical distributions. Answer: Thank you for your suggestion. We have changed. For Figure 1, page 31, the color scale goes up to 0.5, while the data is mainly from 0 to 0.3. The authors should consider redoing the graph with a color scale from 0 to 0.3 Answer: Thank you for your suggestion. Figure 1 comes from the 3D gridded level 3 data. We make an integral of altitude samples at each latitude and longitude bin for the CALIOP L3 Ice Cloud product, transforming ICF from 3D into 2D. And we deleted the Fig.1 and 2 because we changed the study content and title, now paid mainly attention to vertical distribution of diurnal variation of ice cloud properties. For Figure 2, page 31, the color scale goes up to 0.01, while the data is mainly from 0 to 0005. The authors should consider redoing the graph with a color scale from 0 to 0.005 Answer: Thank you for your suggestion. Figure 2 comes from the 3D gridded level 3 data. We make an integral of altitude samples at each latitude and longitude bin for the CALIOP L3 Ice Cloud product, transforming ICF from 3D into 2D. And we deleted the Fig.1 and 2 because we changed the study content and title, now paid mainly attention to vertical distribution of diurnal variation of ice cloud properties. Line 232, page 11: Use the same g/m3 units in the text as in Figure 5. Answer: Thank you for your suggestion. We have corrected it. Line 235, page 11: I did not understand what the "spike-shaped structure" refers to. From previous CALIPSO papers, this structure is likely identified with a physical feature. Refer to the literature to

make the structure less mysterious. (Is it related to the melting-band lidar backscatter feature, or something else?) Answer: Thank you for your suggestion. We used the quality flags to re-plot the results and adjust the color range. We find the results still occur the "spike-shaped structures" but the spikes became weaker. So, we also think the spikes possibly are the artifacts based on the retrieved algorithm of IWC. And we have emailed the contact the creator of this product, but without the response. Now, we only point out the spikes are artifact in the paper. Line 238-240, page 11: The sentence is not clear. Please revise. The term "we excluded the maximum" is not clear. Answer: We excluded the maximum of the IWC (in Fig.5 that we re-plot, the small red rectangle contains one IWC data over the SH polar region in summer, the big red rectangle zooms in on the small red rectangle) because the value lager than 0.01g/m3. Line 328, page 15: change to during night compared to day. Answer: Thank you for your suggestion. We have changed it. In Figure 8 (page 37), it may be better to graph 100 (# night observations – # day observations) / # night observations instead of # night observations - # day observations. Answer: Thank you for your suggestion. We have changed it. Line 387, page 18: The phrase "the values of these parameters were obtained from the CALIPSO platform" implies that the RH and temperature profiles are measured by the CALIPSO experiment. It would be better to state that auxiliary files specify these profiles. Please specify the origin of the auxiliary files. Answer: Thank you for your suggestion. According to the opinions of other experts, we have changed the study content and title, now paid mainly attention to vertical distribution of diurnal variation of ice cloud properties. So, we dropped the section. Line 388-389, page 18: change to Fig. 9 shows the 10-year global contour density plots of nighttime and daytime IWC, RH, and TE. Answer: Thank you for your suggestion. According to the opinions of other experts, we have changed the study content and title, now paid mainly attention to vertical distribution of diurnal variation of ice cloud properties. So, we dropped the Fig.9. Line 443-446, page 20: Rephrase. See comment on lines 206-209. Answer: Thank you for your suggestion. We have re-written it. Table 3. The numbers are too small. Expand the table into two parts to increase the font size. Answer: Thank you

for your suggestion. According to the opinions of other experts, we have changed the study content and title, so now paid mainly attention to vertical distribution of diurnal variation of ice cloud properties. So, we dropped the Table 3.
* * *